# TOWARDS A PHYSICS FOUNDATION MODEL

## ABSTRACT

Foundation models have revolutionized natural language processing through a "*train once, deploy anywhere*" paradigm, where a single pre-trained model adapts to countless downstream tasks without retraining. Access to a **Physics Foundation Model (PFM)** would be transformative—democratizing access to high-fidelity simulations, accelerating scientific discovery, and eliminating the need for specialized solver development. Yet current physics-aware machine learning approaches remain fundamentally limited to single, narrow domains and require retraining for each new system. We present the **General Physics Transformer (GP$_{hy}$T)**, trained on 1.8 TB of diverse simulation data, that demonstrates foundation model capabilities are achievable for physics. Our key insight is that transformers can learn to infer governing dynamics from context, enabling a single model to simulate fluid-solid interactions, shock waves, thermal convection, and multi-phase dynamics without being told the underlying equations. GP$_{hy}$T achieves three critical breakthroughs: (1) superior performance across multiple physics domains, outperforming specialized architectures by more than 7x, (2) plausible zero-shot generalization to entirely unseen physical systems through in-context learning, and (3) more stable long-term predictions through long-horizon rollouts. By establishing that a single model can learn generalizable physical principles from data alone, this work opens the path toward a universal PFM that could transform computational science and engineering.

## 1 INTRODUCTION

Over the last few years, massive accumulation of data and large-scale GPU computing have led to extraordinary advancements in language model (LLMs) capabilities (Devlin et al., 2019; Radford et al., 2018; 2019; Raffel et al., 2020; Anil et al., 2023). These frontier foundation models, often exceeding 100 billion parameters, have established a "*train once, deploy anywhere*" paradigm. They generalize to unseen domains and can be prompted to perform diverse tasks—from coding to creative writing—without task-specific finetuning (Brown et al., 2020; Minaee et al., 2025), exhibiting emergent abilities not explicitly programmed (Wei et al., 2022).

We envision a similar paradigm for physics-aware machine learning (PAML), where a single foundation model can simulate a wide range of physical systems, boundary conditions, and initial states. This allows end-users to employ the model for their individual use case without the need for extensive retraining or fine-tuning. However, current state-of-the-art physics models, such as physics-informed neural networks (PINNs) (Raissi et al., 2019), neural operators (Kovachki et al., 2023; Lu et al., 2021), and physics-aware recurrent convolutions (PARC) (Nguyen et al., 2023a; 2024), are fundamentally limited to solving a single, narrowly scoped physical system: While they excel at their specific task, they cannot generalize to new physics or boundary conditions without new or additional training. Even recent multi-physics approaches largely rely on fine-tuning or meta-learning, which still demands new data and training for each application (Penwarden et al., 2023; Cho et al., 2023; McCabe et al., 2024; Cao et al., 2025; Hao et al., 2024; Herde et al., 2024; Morel et al., 2025).

The primary barrier to a truly "*train once, deploy anywhere*" Physics Foundation Model (PFM) is the immense diversity of physical phenomena coupled with highly expensive and limited data. A single model intended as a surrogate for computational fluid dynamics, for example, must reconcile the micrometers and milliseconds of microfluidics with the kilometers and hours of weather forecasting. Critically, the same initial state can evolve into vastly different outcomes depending on the governing physical laws. A PFM must therefore either be explicitly provided with a complete system description

(e.g. scale, boundary conditions, material properties, governing equations) or infer the dynamics from the given input data itself. As the variety of problems scales, the former becomes impractical and defeats the purpose of a truly general model.

Here, we propose that the path towards PFMs lies in emulating the in-context learning abilities of LLMs (Brown et al., 2020; Agarwal et al., 2024). Instead of being explicitly told the governing equations, a model should infer the underlying dynamics from a "prompt" consisting of a short sequence of prior states. Such a model could adapt its predictions on the fly, enabling a single, unified architecture to tackle a wide array of physical scenarios. Furthermore, the model should use little implicit bias to remain as general as possible. This paradigm shift presents fundamental research challenges that we address through the General Physics Transformer (GP$_{hy}$T) trained on a diverse 1.8 TB corpus of simulation data. With this model, we investigate three critical questions:

**Q1:** Can a single, large-scale (but simple) transformer effectively model a wide range of disparate physical systems (e.g., incompressible flow, shock waves, convection)? (Section 4.1)

**Q2:** Can GP$_{hy}$T maintain physical consistency and stability during extended autoregressive rollouts, a characteristic crucial for real-world application? (Section 4.2)

**Q3:** Can this foundation model perform zero-shot generalization to new, unseen physical conditions (e.g., new boundary conditions, entirely new physics) by inferring the dynamics from the input alone? (Section 4.3)

Our results demonstrate that GP$_{hy}$T not only outperforms specialized architectures on seen tasks but also successfully generalizes to out-of-distribution problems, including producing physically plausible predictions for phenomenon absent from its training data. This work represents a critical step towards creating a "universal physics engine" that could democratize access to high-fidelity simulations and accelerate scientific discovery across disciplines.

## 2 RELATED WORK

**Neural Surrogates for Physical Systems** Machine learning has emerged as a powerful tool to accelerate the simulation of complex physical systems, which are governed by partial differential equations (PDEs) that lack analytical solutions. The dominant paradigms in this domain are Physics-Informed Neural Networks (PINNs), Neural Operators (NOs) and their combination Physics-informed Neural Operators. PINNs embed the governing PDEs directly into the training process as a soft constraint in the loss function, which enhances data efficiency and physical consistency (Raissi et al., 2019; Karniadakis et al., 2021). This approach has been successfully applied across numerous scientific fields (Faroughi et al., 2024). Neural Operators, in contrast, learn the solution operator mapping from the PDE parameters to the solution space, making them discretization-invariant (Kovachki et al., 2023). Prominent examples include Fourier Neural Operators (FNOs), which perform convolutions in the frequency domain (Li et al., 2020), and DeepONets (Lu et al., 2021). Moreover, combinations of operators with physics-informed loss functions can reduce the data requirements of neural operators (Goswami et al., 2023; Li et al., 2023).

Despite their success, PINNs and NOs are fundamentally specialized solvers. They are typically designed and trained for a single, well-defined physical system and struggle to generalize to new governing equations, boundary conditions, or complex multi-physics phenomena without transfer learning (Goswami et al., 2022; 2020) or full retraining. This inherent specialization prevents them from serving as true "foundational" models in the way that large language models (LLMs) do for natural language tasks.

**Towards Foundational Models for Science** The concept of a large-scale foundation model pre-trained on extensive, diverse data has begun to permeate scientific disciplines. This has led to two distinct categories of models. The first involves language-based models fine-tuned on scientific corpora, such as AstroLLaMA for astronomy (Nguyen et al., 2023b) or specialized models for interpreting medical records (Jiang et al., 2023). The second category comprises models that operate directly on quantitative scientific data such as velocity or temperature fields. Notable examples are models for molecular structures (Chithrananda et al., 2020), climate forecasts (Nguyen et al., 2023c), or aquatic science (Yu et al., 2025). Regardless of methodology, any foundation model must either be capable generalizing to unseen data or finetuned for new tasks (Choi et al., 2025).

In physics, the pursuit of foundation models has largely focused on enhancing the generalization of neural surrogates. Researchers have explored meta-learning (Penwarden et al., 2023; Morel et al., 2025) and transfer learning (Subramanian et al., 2023; Goswami et al., 2022) to adapt pretrained models to new PDE systems with fewer data samples. Recently, multi-tasks models trained on multiple physical systems were explored. These models either use multiple (McCabe et al., 2024; Hao et al., 2024; Cao et al., 2025; Nguyen et al., 2025) or single time steps (Herde et al., 2024) as input and autoregressively predict the temporal evolution of the system. However, while these groups demonstrated superior accuracy compared to single-physics models, all opted for finetuning to unseen tasks. These efforts represent important progress, but they still fall short of the "*train once, deploy anywhere*" paradigm that we envision.

**Transformers for Spatiotemporal Modeling**   The architectural backbone of most modern foundation models is the Transformer (Vaswani et al., 2017), whose self-attention mechanism has proven exceptionally effective at capturing long-range dependencies in sequential data. Originally developed for language, this architecture was successfully adapted for computer vision in the Vision Transformer (ViT) (Dosovitskiy et al., 2020). By treating an image as a sequence of patches, ViTs achieved state-of-the-art performance with sufficient data provided (Khan et al., 2023). This concept was further extended to video by creating spatiotemporal "tubelet" tokens (Arnab et al., 2021), enabling transformers to model dynamic visual data. The power of transformers also extends to generative tasks. Using vector quantization, auto-regressive transformer models can operate on a discrete latent space of visual tokens (Esser et al., 2021; Chang et al., 2022; Ramesh et al., 2021).

# 3 GENERAL PHYSICS TRANSFORMER

## 3.1 ARCHITECTURE

Due to data scarcity, today's physics models must incorporate inductive biases for optimal performance. However, the diversity of multiple physical systems restricts such choices. The General Physics Transformer ($\text{GP}_{\text{hy}}\text{T}$) is designed as a hybrid model that integrates a deep learning component within a classic numerical methods framework. As illustrated in Figure 1a, the core of our architecture is a Transformer-based neural differentiator that learns the temporal dynamics of a system, coupled with a standard numerical integrator that extrapolates the system's future state. This approach, inspired by Neural ODEs (Chen et al., 2018) and previous work of Nguyen et al. (2024), allows the model to predict the evolution of diverse physical systems governed by partial differential equations (PDEs).

**Neural differentiator**   The neural differentiator (blue dashed box) models the partial derivative ($\frac{\partial X}{\partial t}$) of the physical state with respect to time. $X$ is composed of multiple physical fields (channels), such as pressure, temperature, and velocity. To allow for in-context learning, the differentiator receives multiple time snapshots ($X_{t_i-n}, ..., X_{t_i}$) of the physical state. The sample is then tokenized by a single linear transformation across spatial and temporal dimensions, yielding non-overlapping spatiotemporal (tubelet (Arnab et al., 2021)) patches. The size of these patches control the number of spatial and temporal pixels encoded in each token. Absolute positional encodings are added to the patches. The spatiotemporal transformer consists of multiple transformer layers with layer norms and attention across all time and space dimensions, illustrated in Figure 1b. We chose this unified attention mechanism over more computationally efficient factorized approaches to ensure maximum expressivity, allowing the model to capture complex, non-separable phenomena like turbulence and shockwave interactions. Finally, a linear transformation (detokenizer) reverts the spatiotemporal patches into the input space.

To provide the model with explicit local information, we compute the first-order spatial ($dx, dy$) and temporal ($dt$) derivatives of the input fields using central differences. These computed derivatives are concatenated with the original fields along the channel dimension, enriching the input for the neural differentiator. This technique is particularly effective for resolving phenomena with sharp gradients (Cheng et al., 2024).

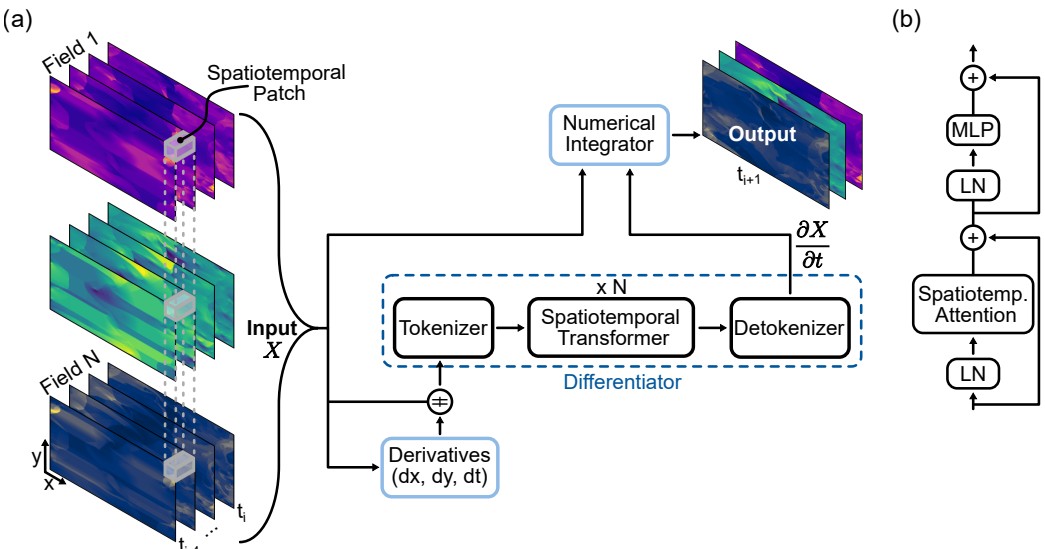

Figure 1: (a) General architecture of GP$_{hy}$T. A 4D-stack of physical quantities (time, height, width, fields) serves as input $X$. The numerically computed derivatives of each field are concatenated to the input. The differentiator (linear tokenizer, spatiotemporal transformer, linear detokenizer) provides the partial derivative of $X$ wrt. time. Finally, a numerical integrator computes the next timestep of each field given $\frac{\partial X}{\partial t}$ and $X$. (b) Architecture of a single transformer layer, consisting of layer norms (LN), spatiotemporal attention, and multilayer perceptron (MLP).

**Numerical Integrator**   With the learned time derivative, we can predict the next state of the system, $X_{t_{i+1}}$, using a numerical integration step. The general form of the integration is:

$$X_{t_{i+1}} = f \left( X_{t_i}, \left.\frac{\partial X}{\partial t}\right|_{t_i}, \Delta t \right) \tag{1}$$

In this study, we choose the first-order Forward Euler method, however more accurate integrators like Heun or Runge–Kutta 4 can be chosen as well.

### 3.2   DATASETS

To train a model capable of learning general physical principles, we curated a large and diverse corpus of simulation data, comprising seven distinct datasets listed in Table 1. The combined dataset contains over 2.4 million simulation snapshots, totaling 1.8 TB of data. Our data is sourced from both the publicly available "The Well" benchmark (Ohana et al., 2024) and our own custom simulations, described in detail in Appendix 6.3.

The three datasets from The Well cover a range of fundamental physics, including incompressible (Shear flow) and compressible (Euler) fluid dynamics, and thermal convection (Rayleigh–Bénard). However, these systems largely lack the solid boundaries and complex geometries prevalent in engineering applications. To address this, we generated four additional datasets featuring flows around rigid obstacles, Rayleigh–Bénard with additional obstacles, heat exchange with solid elements (Thermal Flow), and multi-phase dynamics in porous media. These additions introduce critical physical behaviors, such as boundary layer formation, vortex shedding, pressure-driven instabilities, as well as varying physical scales, significantly expanding the diversity of the training data.

A core objective of this work is to train a model that can generalize by inferring the underlying physics from context. To facilitate this, we implemented two crucial data augmentation strategies:

- **Variable Time Increments:** Each simulation trajectory is sub-sampled using multiple time-step increments ($\Delta t$). This forces the model to learn dynamics that are invariant to the sampling

Table 1: Dataset overview. Each trajectory has different initial conditions / randomized geometry and contains the given number of timesteps. Unique samples are all possible combinations of 4 input and 1 output snapshots sampled for time-increments of 1-8 and random axis flips.

| Dataset | Trajectories | Timesteps | Snapshots | Unique samples | Origin |
|---|---|---|---|---|---|
| Shear flow | 1120 | 200 | 224,000 | 6,522,880 | The Well |
| Rayleigh–Bénard | 1750 | 200 | 350,000 | 10,192,000 | The Well |
| Euler | 5000 | 100 | 500,000 | 13,120,000 | The Well |
| Obstacle flow | 1266 | 481 | 608,946 | 18,756,856 | Own |
| Thermal flow | 354 | 481 | 170,274 | 5,244,864 | Own |
| Rayleigh–Bénard 2 | 228 | 1001 | 228,228 | 7,171,968 | Own |
| Twophase flow | 816 | 401 | 327,216 | 10,000,896 | Own |
| **Sum** | | | **2,408,664** | **71,009,464** | |

frequency. For any given input, the model must infer the temporal scale from the dynamics presented in the prompt, as a single time step could represent milliseconds in one context and minutes in another.

- **Per-Dataset Normalization:** The physical phenomena in our corpus span vastly different scales, from micrometer-sized pores in two-phase flow to large-scale convective cells. To handle this, we normalize each dataset independently. This preserves the relative physical quantities within a single simulation while compelling the model to infer the absolute magnitudes and spatial scales of a new system purely from the context provided by the input snapshots.

By training on this varied data, $GP_{hy}T$ is explicitly pushed to develop in-context learning abilities, rather than memorizing the characteristics of a single, fixed physical system.

## 4 RESULTS

### 4.1 MULTI-PHYSICS LEARNING

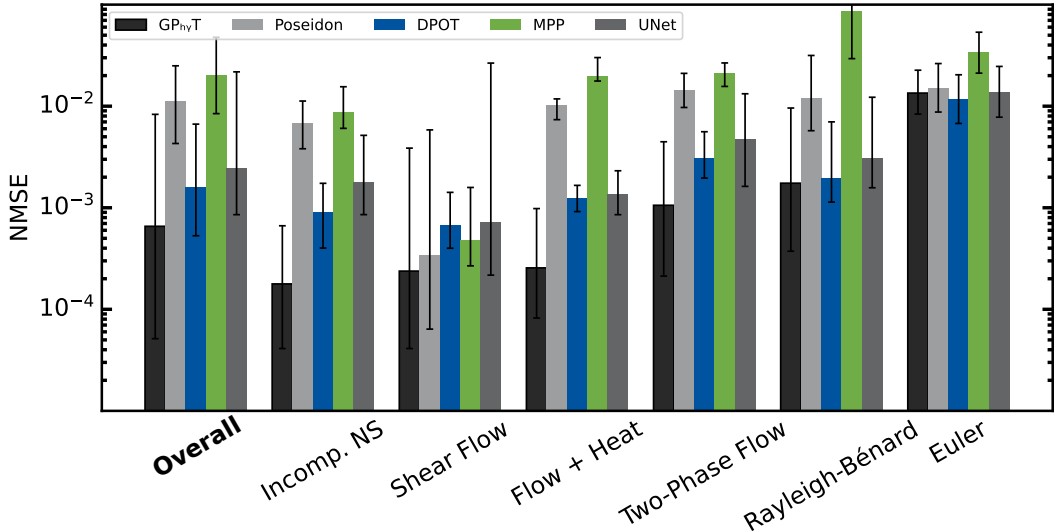

Figure 2: Median normalized mean square error (NMSE) of all models across the test datasets for next step prediction. Losses are grouped for each dataset and the overall loss. The error bars indicate the 25th and 75th percentile errors. $GP_{hy}T$ shows the overall lowest error and lowest in all but one dataset.

To address our first research question **Q1**—whether a single general model can effectively learn to represent numerous, disparate physical systems—we evaluated $GP_{hy}T$'s single-step prediction accuracy across our entire multi-physics test set. We benchmarked against multiple established baseline models designed for multi-physics predictions: MPP (McCabe et al., 2024), DPOT (Hao et al., 2024), and Poseidon (Herde et al., 2024). All three are transformer-based architectures that have demonstrated state-of-the-art performance on learning multiple physical systems simultaneously. Both MPP and DPOT use multiple input frames (similar to $GP_{hy}T$), while Poseidon uses only a single input timestep. Additionally, we included a standard UNet as a baseline without implicit physics-specific biases. All models were trained identically to predict the subsequent frame, ensuring a fair comparison.

Figure 2 presents the median normalized mean squared error (NMSE) for all test datasets as well as the overall error. We report the median rather than the mean, as all models experience occasional severe mispredictions that heavily skew the mean error, likely caused by the normalization of the error and select few extremely challenging samples. Overall, $GP_{hy}T$ demonstrates substantial accuracy gains over all baselines, with the next-best model (DPOT) exhibiting $7\times$ higher NMSE. Notably, the UNet also achieves lower error than the more sophisticated multi-physics models Poseidon and MPP.

Examining the per-dataset results reveals that different physical systems present vastly different challenges. Incompressible, steadily moving fluids (incompressible Navier-Stokes and shear flow) yield the lowest errors across all models. More complex systems involving heat transfer, such as Rayleigh-Bénard convection, or the Euler dataset, which features shockwaves and sharp discontinuities, present greater challenges and result in higher errors. This is expected, as such phenomena are notoriously difficult to resolve accurately. Nevertheless, $GP_{hy}T$ demonstrates robust performance across all tasks, achieving the lowest NMSE on all physics except Euler. While the three leading models ($GP_{hy}T$, UNet, and DPOT) show a steady increase in NMSE for more challenging tasks, the NMSE for Poseidon and MPP remains nearly constant – potentially indicating difficulties in adapting to varying systems. It is important to note that all models still exhibit high variability in their prediction accuracy, as illustrated by the large error bars. These results suggest that the $GP_{hy}T$ architecture, despite its generality and simplicity, is inherently better suited as a Physics Foundation Model than other multi-physics approaches. Its ability to dynamically attend to relevant spatiotemporal features appears to be a key advantage for handling diverse physical phenomena within a single model.

## 4.2 LONG-RANGE PREDICTION

As stated in research question **Q2**, the true utility of any physics surrogate model is measured by its ability to maintain stability and accuracy over extended temporal horizons. This task is exceptionally challenging **and remains unsolved**, as it requires the model to generate a full trajectory from an initial state, with prediction errors from each step accumulating over time. Long-range prediction constitutes a critical test of a model's physical consistency and represents a common failure point that is often omitted in the literature.

Figure 3a shows the median NMSE as a function of autoregressive prediction steps. During the initial phase of the rollout ($t = 1$–$8$), all models exhibit significant error accumulation, which continues at a similar rate for longer time horizons. Interestingly, Poseidon shows less initial error accumulation but starts from already elevated error levels. Compared to the baselines, $GP_{hy}T$ performs exceptionally well, only surpassed slightly by Poseidon in the later stages of the rollout. Inspection of 12-step and 24-step rollout NMSE for individual datasets (see Appendix 6.1) reveals that Poseidon and $GP_{hy}T$ are competitive across most datasets, with the exception of shear flow, where Poseidon achieves substantially lower error rates, albeit with significant error bars. Since the slopes are similar across all models, we hypothesize that the initial error is of critical importance. Furthermore, error accumulation appears to be an inherent property of all neural architectures, suggesting that hybrid models or alternative approaches may be necessary to mitigate this limitation.

To investigate the failure modes of the models, we visualized representative prediction samples from the best-performing models as well as the worst (MPP) in Figure 3b and Appendix 6.4. Upon close inspection, Poseidon is unable to reproduce high-frequency details such as Euler shockwaves, whereas $GP_{hy}T$ successfully preserves the global dynamics and physical plausibility of the flow. MPP and DPOT, which perform worst in autoregressive rollouts, often fail to match the ground truth data

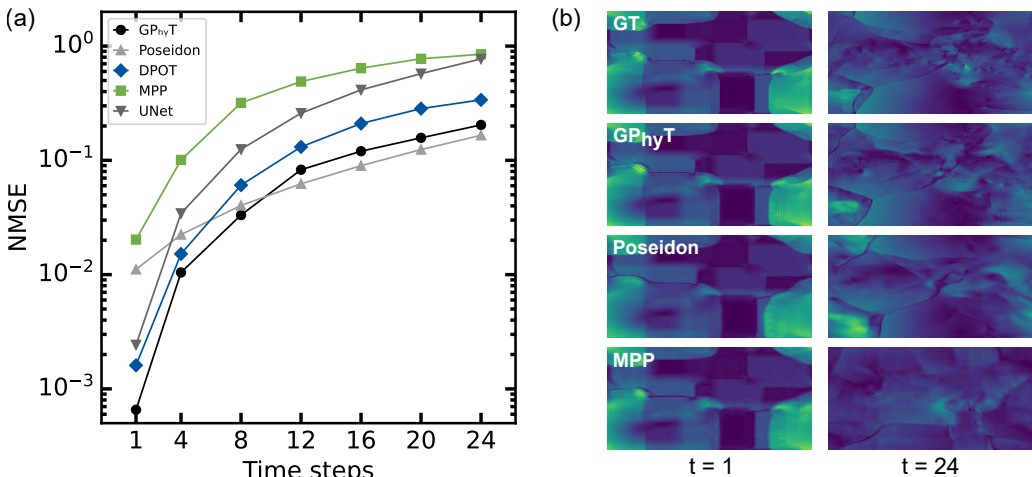

Figure 3: (a) Overall autoregressive long-horizon prediction (median NMSE) for all models on the known physical systems up to 24 prediction steps. (b) Visualization of prediction step t=1 and t=24 of Euler shockwaves with ground truth (GT), the worst model (MPP) and the two best (Poseidon, GP$_{hy}$T). The images can be best viewed on a high-definition digital monitor.

entirely. Overall, it must be emphasized that no model is capable of high-fidelity predictions, let alone replacing numerical solvers. However, GP$_{hy}$T shows not only the highest accuracy but also the most physical plausible and consistent predictions. Achieving the precision required for practical engineering applications will necessitate orders of magnitude improvement in both accuracy and stability.

### 4.3 IN-CONTEXT LEARNING

The defining characteristic of a true foundation model is its ability to adapt to new tasks without additional training – a capability that fundamentally distinguishes foundation models from traditional specialized approaches. In language models, this emerges through in-context learning, where models leverage prompts to perform tasks never explicitly seen during training (Brown et al., 2020). To investigate our research question **Q3**, whether GP$_{hy}$T exhibits similar emergent capabilities for physics, we designed increasingly challenging generalization experiments. First, we evaluated the model on systems with modified boundary conditions that were completely absent from the training data. Second, we pushed the boundaries further by presenting entirely novel physical phenomena, including supersonic flows and turbulent radiative layers never encountered during training. These experiments probe whether the model has learned transferable physical principles rather than merely memorizing dataset-specific patterns.

The quantitative results are summarized in Figure 4. Additionally, predictions from all models are visualized in Appendix 6.5. Overall, GP$_{hy}$T outperforms all other foundation models on the crucial long-horizon predictions. Notably, it is the only model that remains below an NMSE of 1, which is the threshold beyond which predicting the simple mean of the image would yield higher accuracy. The visualizations in the Appendix show that GP$_{hy}$T and UNet, as the more general models, maintain greater stability and avoid diverging predictions. Figure 4b presents the same data split into systems with novel boundary conditions (top) and completely novel physical systems (bottom). For incompressible flow around obstacles and the complex Euler shockwaves with new boundary conditions, our model significantly outperforms the baselines. This demonstrates that GP$_{hy}$T successfully infers new boundary conditions from the prompt alone, without requiring fine-tuning.

The most challenging tests involve two completely new physical systems: supersonic flow around an obstacle and a turbulent radiative layer. Here, the initial NMSE is considerably higher than for the known systems, underscoring the significant challenge posed by truly out-of-distribution physics. Nevertheless, Figure 4b shows that GP$_{hy}$T still achieves lower errors than all baselines. Fine-tuning the model on these new systems, as done in previous studies, would naturally yield even

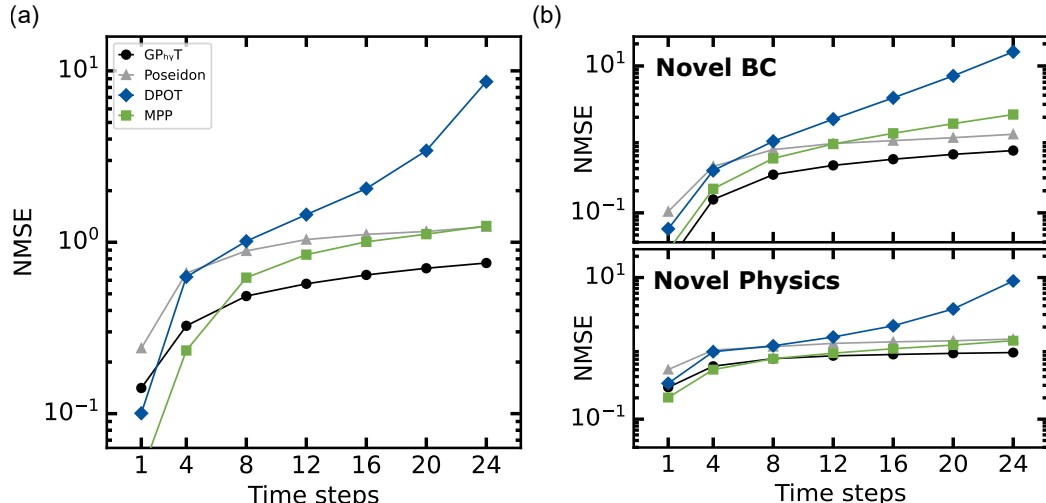

Figure 4: (a) Overall autoregressive long-horizon prediction (median NMSE) for all models on the novel physical systems up to 24 prediction steps. (b) Separate graphs for autoregressive long-horizon prediction of systems with new boundary condations (top) and completely new physics (bottom).

more accurate predictions (McCabe et al., 2024; Herde et al., 2024; Hao et al., 2024). However, the ability to extrapolate and produce physically plausible results for entirely new physics—even with reduced accuracy—constitutes a powerful demonstration of emergent generalization. This capability represents a fundamental step toward the "train once, deploy anywhere" paradigm that defines true foundation models.

## 4.4 ABLATION: WHAT MAKES A GOOD PHYSICS FOUNDATION MODEL

To understand the key design choices that enable $GP_{hy}T$'s strong performance, we conduct ablation studies examining architectural components, model scale, and input context length. Two design choices distinguish $GP_{hy}T$ from a standard video transformer: predicting the time derivative $\frac{\partial X}{\partial t}$ rather than the next state directly, and providing explicit spatial and temporal derivatives as additional input features. Figure 5a compares our neural differentiator framework against direct next-state prediction, showing that decoupling the learning of dynamics from numerical time integration yields substantially lower errors across all rollout horizons. We hypothesize that predicting derivatives provides a more natural learning target that generalizes better across different temporal scales. Figure 5b demonstrates the impact of explicit derivative features: without the computed spatial ($\partial_x$, $\partial_y$) and temporal ($\partial_t$) derivatives concatenated to the input, prediction accuracy degrades by nearly an order of magnitude for extended rollouts, as these features provide crucial local gradient information for resolving sharp gradients and discontinuities. Together, these results show that a general-purpose video transformer, augmented with lightweight physics-motivated modifications, can outperform specialized multi-physics architectures.

Beyond architectural choices, $GP_{hy}T$ exhibits favorable scaling behavior (Figure 5c): increasing model capacity from 9M to 385M parameters yields consistent accuracy improvements, suggesting that further gains may be achievable with larger models. Finally, the number of input timesteps ($N_{input}$) directly controls the model's capacity for in-context learning. Figure 5d shows that a single input timestep, which provides no temporal context, produces the highest errors, while the largest performance gain occurs when increasing from one to two timesteps. Additional timesteps continue to improve accuracy with diminishing returns, following a log-linear trend. This analysis has practical implications: while longer prompts improve accuracy, they increase computational cost quadratically due to self-attention and must be generated by a numerical solver at inference time. Our results suggest that $N_{input} = 4$ provides a favorable accuracy-efficiency trade-off, requiring substantially fewer input frames than prior work (McCabe et al., 2024; Hao et al., 2024) while maintaining competitive performance.

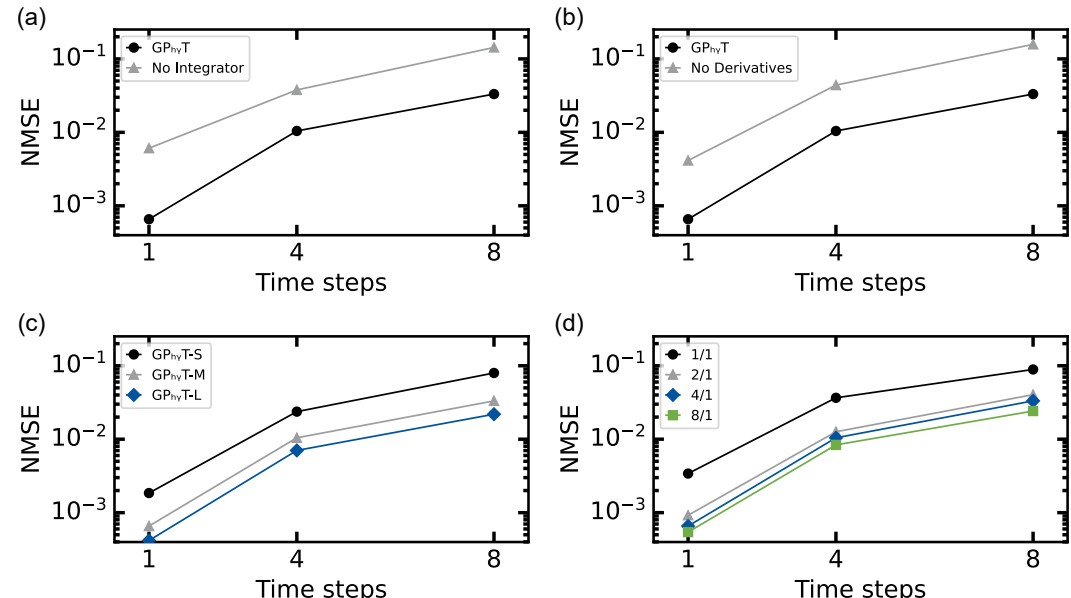

Figure 5: Ablation studies on the known datasets with median NMSE: (a) Comparing GP$_{hy}$T against a version predicting the next state directly. (b) Comparing GP$_{hy}$T against a version without explicit spatial and temporal derivatives as input. (c) Scaling behavior of GP$_{hy}$T. (d) Effect of the number of input time steps on rollout performance.

## 4.5 LIMITATIONS

While GP$_{hy}$T demonstrates promising advances toward true physics foundation models, several key limitations remain:

**2D data constraints:** Due to data scarcity and computational limitations, the current model, as well as most other multi-physics models are restricted to 2D systems. However, the proposed architecture is directly extensible to 3D systems, and the increased computational demands can be mitigated by employing larger temporal patch sizes.

**Long-term stability:** Although GP$_{hy}$T achieves remarkable accuracy in long-term rollout predictions, it falls considerably short of the precision exhibited by numerical solvers. Significantly lower prediction errors are essential for practical engineering applications.

**Limited physics coverage:** GP$_{hy}$T is currently trained exclusively on fluid dynamics and heat transfer systems. A comprehensive physics foundation model would require incorporation of diverse physical domains, including mechanics, chemistry, molecular dynamics, and optics.

**Fixed domain resolution:** The model is trained on 256×128 resolution images. While this resolution is adequate for many simulation scenarios, widespread adoption may necessitate a model capable of handling variable domain sizes and resolutions. However, per-dataset normalization allows GP$_{hy}$T to train on multiple resolutions of the same system, effectively learning discretization-invariance.

## 5 CONCLUSION

We have demonstrated that a simple, general transformer-based model can effectively learn and predict the dynamics of diverse physical systems without explicit physics-specific features, marking a significant step toward true Physics Foundation Models. GP$_{hy}$T not only outperforms other state-of-the-art multi-physics model on known physics but, more importantly, exhibits emergent in-context learning capabilities—inferring new boundary conditions and even entirely novel physical phenomena from input prompts alone. This "*train once, deploy anywhere*" capability, previously exclusive to language models, opens new possibilities for physics simulation. Our hybrid architecture,

combining a transformer-based neural differentiator with numerical integration, proves that the attention mechanism can capture complex spatiotemporal dependencies across vastly different scales. The model's ability to maintain physical consistency through long-horizon rollouts, while not yet matching numerical solvers, demonstrates that learned representations can encode generalizable physical principles rather than merely memorizing dataset-specific patterns. The path toward a comprehensive Physics Foundation Model requires addressing current limitations: extending to 3D systems, incorporating diverse physical domains beyond fluid dynamics, and achieving variable-resolution capabilities. Most critically, improving long-term stability will be essential for practical engineering applications. Nevertheless, $GP_{hy}T$ establishes that the foundation model paradigm, i.e. a single pre-trained model adapting to novel tasks through context alone, is achievable for physics. As we scale both model capacity and training data diversity, we anticipate further emergent capabilities that could fundamentally transform how we approach computational physics, making high-fidelity simulations accessible to researchers and engineers without the traditional barriers of specialized solver development or extensive computational resources.

## REPRODUCIBILITY STATEMENT

All training, evaluation and plotting code will be published. Furthermore, model checkpoints will be released on Hugging Face. Finally, the datasets which are not available on The Well will be available to download.

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

# 6 APPENDIX

## 6.1 LONG-HORIZON ACCURACY PER DATASET

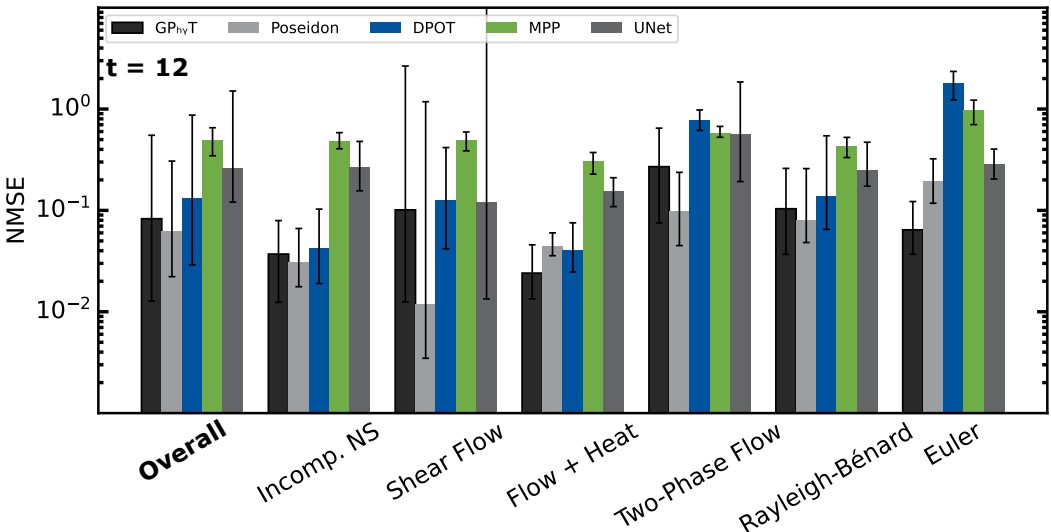

Figure 6: Median normalized mean square error (NMSE) of all models across the test datasets for 12 steps of rollout. Losses are grouped for each dataset and the overall loss. The error bars indicate the 25th and 75th percentile errors.

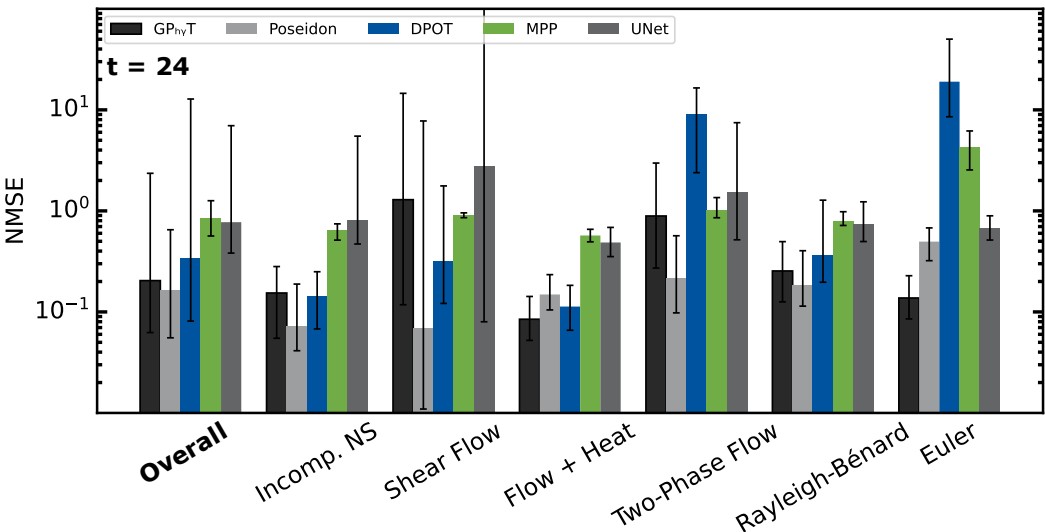

Figure 7: Median normalized mean square error (NMSE) of all models across the test datasets for 24 steps of rollout. Losses are grouped for each dataset and the overall loss. The error bars indicate the 25th and 75th percentile errors.

## 6.2 MODEL & TRAINING HYPERPARAMETERS

### 6.2.1 GENERAL PHYSICS TRANSFORMER

For the models S, M, and L, we train on 4 Nvidia H100 or A100-80GB in parallel. The models are trained for 1 million optimizer steps. We always use a combined batch size of 256. Due to the size

Table 2: GP$_{hy}$T parameters with number of transformer layers, size of embedding (patch) dimension, size of MLP dimension, and number of heads in the multi-head attention. Tflops represent the number of teraflops per update (forwards & backwards).

| Model | #params [M] | Num layers | Embedding dim | MLP dim | Num heads | Tflops |
|---|---|---|---|---|---|---|
| S | 9.2 | 12 | 192 | 768 | 3 | 123.52 |
| M | 112 | 12 | 768 | 3072 | 12 | 698.56 |
| L | 385 | 24 | 1024 | 4096 | 16 | 1,567.24 |

Table 3: Training and model hyperparameter

| | |
|---|---|
| Pos. encodings | absolute |
| Activation function | GELU |
| Norm | Layer norm |
| Optimizer | AdamW ($\beta_1 = 0.9, \beta_2 = 0.999$) |
| Learning rate | lin. warmup (5K steps), |
| | then cosine decay to 1e-6 |
| Batch size | 256 |
| Precision | bfloat16 |
| Gradient norm ($L^2$) | 1 |
| Dropout | no |
| Stochastic depth (Huang et al., 2016) | no |

of each sample (4-dimensional), at minimum 16 dataloader workers are used to fetch the samples. Training and evaluation was done with Pytorch 2.7. We used linear warmup of the learning rate over 5000 steps to 1e-4. After that, a cosine decay schedule with a final learning rate of 1e-6 was used. To stabilize the training, we employ gradient normalization using $L^2$ norm equal to 1. A complete list of model parameter is given in Table 2 and additional hyperparameters are given in Table 3. Teraflops are calculated for a full (multi-gpu) batch with the compiled model and bfloat16 using the torchtnt library.

### 6.2.2 REFERENCE MODELS

The Unet model is a standard architecture: Each downsample block doubles the number of channels and halfs the spatial resolution. The upsample block revert this process with skip connections between the corresponding down and upsample blocks. The UNet also receives 4 input time steps to enable spatio-temporal understanding. We employ 2D convolutions and thus the time steps are flattened into the channel dimension. For both models, learning rate and gradient clipping were equal to the GP$_{hy}$T training.

Table 4: UNet model parameters

| Model | Parameters [M] | Num down/up blocks | Hidden dim at start/end |
|---|---|---|---|
| Unet-M | 124 | 4 | 64 |

All baseline models were trained on 2x A100 GPUs with a total batch size of 256. The models were trained for 1 million steps using the AdamW optimizer. All other parameter were kept true to the respective original publications. Models requiring multiple input time steps used $N_{in} = 4$. Furthermore, all models were trained with the same dataset, including variable strides.

### 6.3 DATASET DETAILS

All datasets used to train the models comprise of a timeseries ($T$) of 2D ($H \times W$) snapshots of a physical domain goverend by common PDE equations such as Navier-Stokes, heat equation or surface tension. Thus, each dataset sample has the form

Table 5: Stats of the reference models.

| Model | Parameters [M] | LR-schedule | LR |
|---|---|---|---|
| DPOT-M (Hao et al. (2024)) | 122 | Cycle | 1e-4 |
| MPP-B (McCabe et al. (2024)) | 116 | warm up & Cosine Decay | 1e-4 |
| Poseidon-B (Herde et al. (2024)) | 156 | warm up & Cosine Decay | 1e-4 |

$$x \in \mathbb{R}^{T \times H \times W \times X} \tag{2}$$

were $X$ are the physical fields, in our case pressure, density, temperature, velocity-x, and velocity-y. Fields not present in the simulation data are provided as zeroed. For training, spatial dimensions of 256 x 128 pixels were used. Datasets with originally larger dimensions were interpolated using bicubic interpolation. Additionally, the Figures 8 and 9 illustrate the general conditions and boundaries of the-well and our simulations, respectively. Each trajectory can be sampled with different $\Delta t$, thus for a given number of snapshots $N_{total}$, a number of input ($N_{in}$) and output snapshots ($N_{out}$) and a given $\Delta t$, $N_{total} - \Delta t(N_{in} + N_{out} - 1)$ unique samples can be generated. Additionally, we employ random axis flips to further increase the diversity of the data. All datasets are split into train/val/test with ratios of 0.8/0.1/0.1.

### 6.3.1 INCOMPRESSIBLE SHEAR FLOW

The shearflow dataset (Ohana et al., 2024) considers a 2D-periodic incompressible shear flow, visualized in Figure 8a). The velocity $\mathbf{u} = (u_x, u_z)$ (horizontal and vertical) and pressure $p$ are governed by the Navier-Stokes equation

$$\frac{\partial \mathbf{u}}{\partial t} - \nu \Delta \mathbf{u} + \nabla p = -(\mathbf{u} \cdot \nabla)\mathbf{u} \tag{3}$$

with the additional constraint $\int p \, dV = 0$ for the pressure gauge. Here, $\Delta = \nabla \cdot \nabla$ is the spatial Laplacian, and $\nu$ is the kinematic viscosity. The shear is initialized by setting the velocity $\mathbf{u}$ in different fluid layers to move in opposite vertical directions. Density and temperature are not considered and thus zeroed in the models input.

### 6.3.2 MULTIQUADRANT EULER

The Euler equations describe inviscid compressible flow governed by

$$\frac{\partial}{\partial t} \iint_{\Omega} U \, dA + \oint_{\partial \Omega} (F\hat{i} + G\hat{j}) \cdot \hat{n} \, dS = 0 \tag{4}$$

$$\tag{5}$$

where

$$U = \begin{pmatrix} \rho \\ \rho u \\ \rho v \\ \rho E \end{pmatrix} \quad \text{and} \quad F = \begin{pmatrix} \rho u \\ \rho u^2 + p \\ \rho u v \\ u(\rho E + p) \end{pmatrix} \quad G = \begin{pmatrix} \rho v \\ \rho u v \\ \rho v^2 + p \\ v(\rho E + p) \end{pmatrix} \tag{6}$$

Here, $t$ is time, $\Omega$ is the control volume with boundary $\partial \Omega$, $A$ is the area, and $S$ is the boundary length. $U$ is the vector of conserved variables, $F$ and $G$ are the flux vectors in the x and y directions respectively, $\hat{i}$ and $\hat{j}$ are the unit vectors in the x and y directions, and $\hat{n}$ is the outward normal vector to the boundary. The conserved variables are density $\rho$, momentum in the x-direction $\rho u$, momentum in the y-direction $\rho v$, and total energy per unit volume $\rho E$, where $u$ and $v$ are the velocity components in the x and y directions, and $E$ is the specific total energy. The pressure is denoted by $p$.

In this dataset (Ohana et al., 2024), the initial pressure field is divided into quadrants with different pressure values, leading to shock waves and other discontinuities. All boundaries are considered as periodic, visualized in Figure 8b). In the original dat, momentum (x,y) was given and thus converted to velocity. Since the system is isothermal, the temperature field is zeroed.

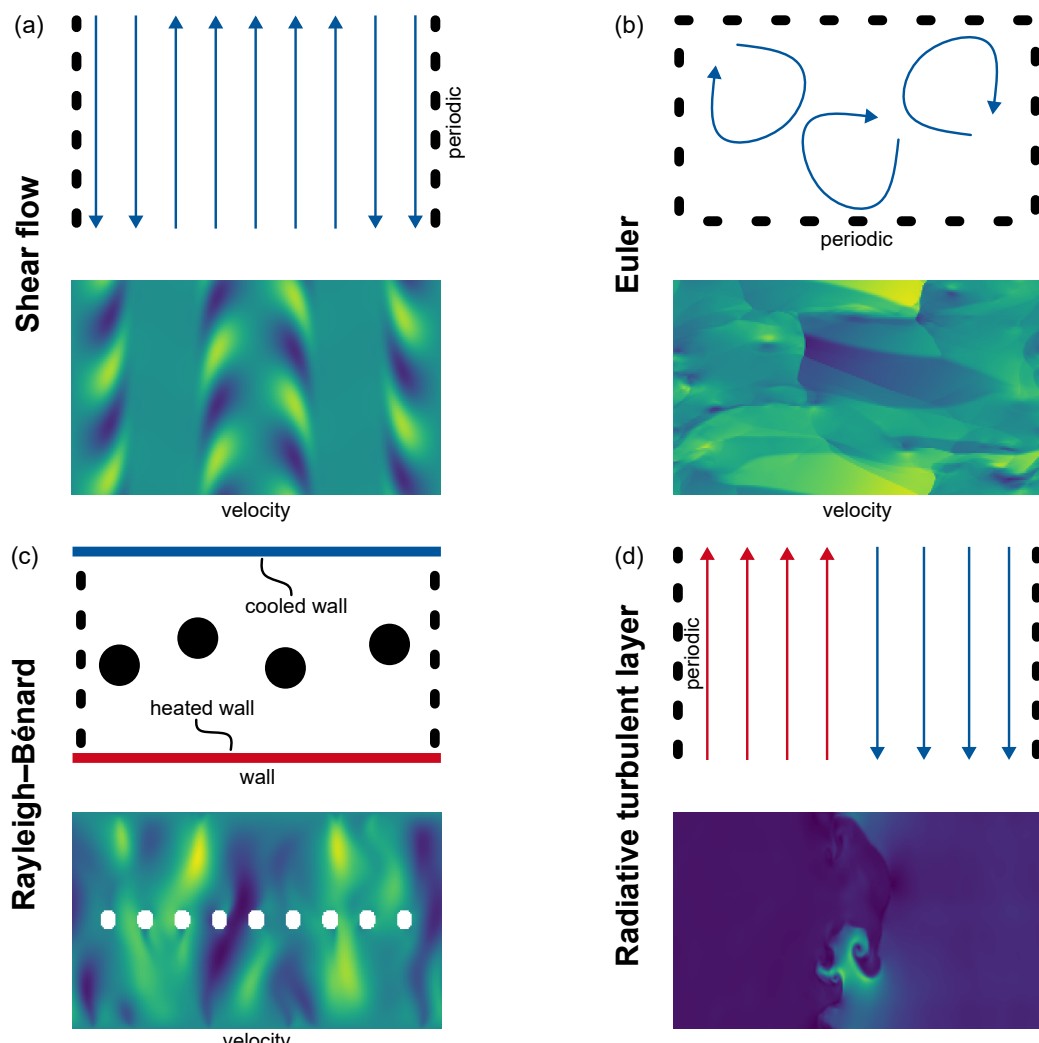

Figure 8: Illustration of physical domain and boundary conditions of the-well datasets. (a) Shearflow with periodic boundary conditions. (b) Simulation of Euler equations initialized with pressure quadrants and periodic boundary conditions. (c) Rayleigh–Bénard convection with heated bottom and cooled top wall, as well as randomly placed obstacles, and periodic boundaries. (d) Turbulent radiative layer with hot and cold gas moving in opposite directions.

### 6.3.3    RAYLEIGH–BÉNARD

Rayleigh–Bénard (Figure 8d) convection occurs between two plates with different temperatures. It is governed by heat transport and fluid flow. Depending on the initial conditions, even tiny variations in temperature or pressure can lead to vastly different fluid behavior. This dataset combines data from the-well (Ohana et al., 2024), which contains no obstacles, and our own data with obstacles.

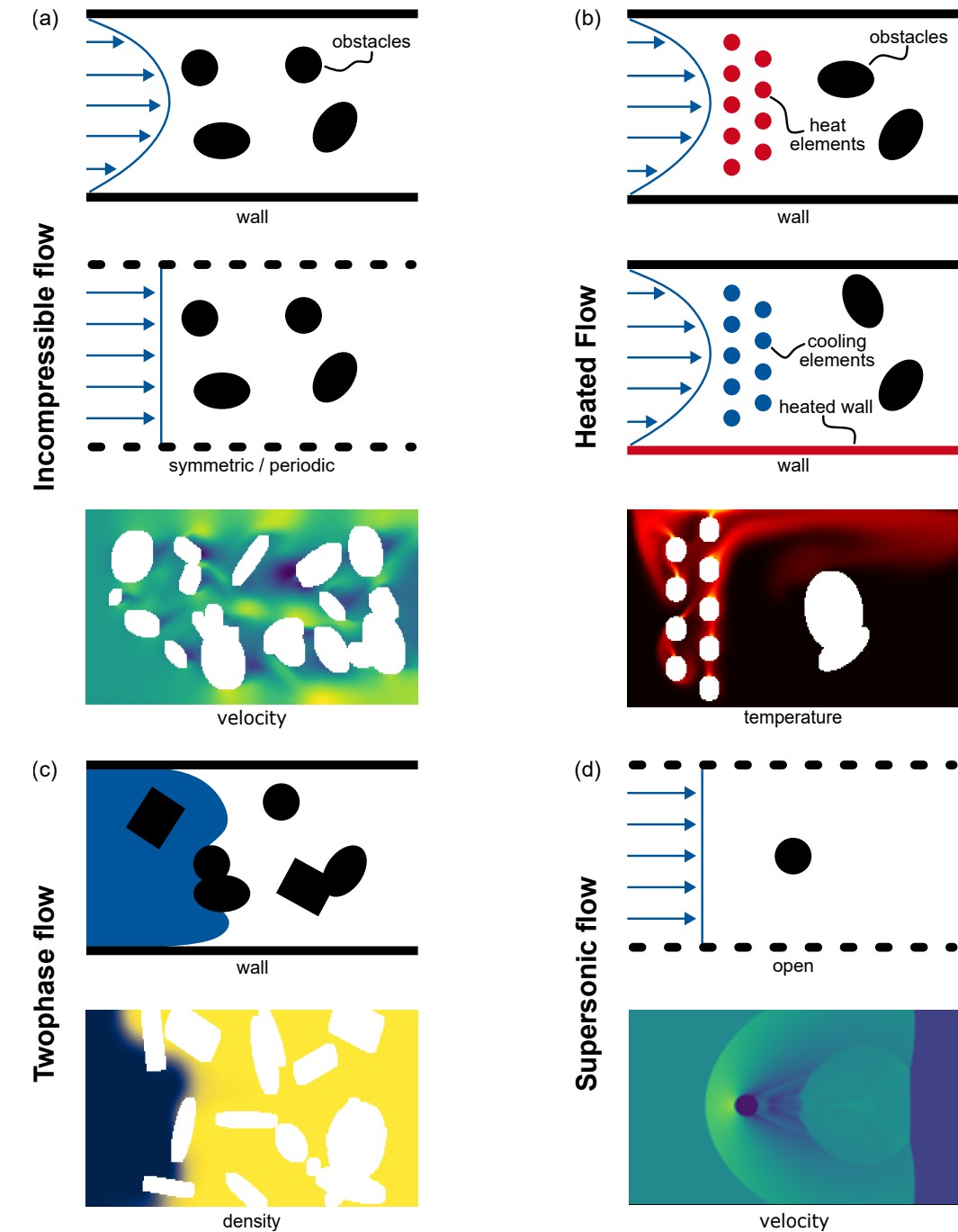

Figure 9: Illustration of physical domain and boundary conditions of our datasets. (a) Incompressible flow around a series of randomly placed obstacles, boundary conditions vary between walls, symmetric, and periodic. (b) Heated flow inside a pipe (walls) with heated elements or walls and isolated obstacles. (c) Twophase flow in random porous media. (d) Supersonic flow with a shock wave hitting a cylinder.

The governing equations are:

$$\frac{\partial b}{\partial t} - \kappa \Delta b = -\mathbf{u} \cdot \nabla b \tag{7}$$

$$\frac{\partial \mathbf{u}}{\partial t} - \nu \Delta \mathbf{u} + \nabla p - b\mathbf{e}_z = -\mathbf{u} \cdot \nabla \mathbf{u} \tag{8}$$

with $\Delta = \nabla \cdot \nabla$ and the constraint $\int p \, dV = 0$. The parameters $\kappa$ and $\nu$ are given by:

$$\kappa = \mathrm{Ra} \times \mathrm{Pr}^{-1/2} \tag{9}$$

$$\nu = \mathrm{Ra}^{1/2} \times \mathrm{Pr}^{-1/2} \tag{10}$$

Here, $b$ represents buoyancy, $\kappa$ is the thermal diffusivity, and $\nu$ is the kinematic viscosity. The velocity vector of the fluid is denoted by $\mathbf{u}$, and $p$ is the pressure. The upward vertical unit vector is given by $\mathbf{e}_z$. The dimensionless parameters governing the system are the Rayleigh number, denoted by Ra, and the Prandtl number, denoted by Pr.

### 6.3.4 TURBULENT RADIATIVE LAYER

The turbulent radiative layer dataset considers a 2D system where hot dilute gas moves relative to cold dense gas, leading to turbulent mixing and radiative cooling processes commonly found in astrophysical environments such as the interstellar and circumgalactic medium, visualized in Figure 8d). This configuration is unstable to the Kelvin-Helmholtz instability, which is seeded with small-scale noise that varies between simulations. The system is governed by the compressible Euler equations with radiative cooling:

$$\frac{\partial \rho}{\partial t} + \nabla \cdot (\rho \mathbf{v}) = 0 \tag{11}$$

$$\frac{\partial (\rho \mathbf{v})}{\partial t} + \nabla \cdot (\rho \mathbf{v}\mathbf{v} + P\mathbf{I}) = 0 \tag{12}$$

$$\frac{\partial E}{\partial t} + \nabla \cdot ((E + P)\mathbf{v}) = -\frac{E}{t_{\mathrm{cool}}} \tag{13}$$

with the equation of state

$$E = \frac{P}{\gamma - 1}, \quad \gamma = \frac{5}{3} \tag{14}$$

Here, $\rho$ is the density, $v = (u, v)$ is the 2D velocity vector, $P$ is the pressure, E is the total energy per unit volume, I is the identity tensor, and $t_{\mathrm{cool}}$ is the cooling time parameter that controls the rate of radiative energy loss.

Initially, cold dense gas is positioned at the bottom while hot dilute gas occupies the top region. Both phases are in thermal equilibrium until mixing occurs, whereupon intermediate temperature gas forms and experiences net cooling, leading to mass transfer from the hot to cold phase. The boundary conditions are periodic in the x-direction with zero-gradient conditions in the y-direction.

### 6.3.5 INCOMPRESSIBLE FLOW WITH OBSTACLES

The dataset include various flow simulations described by the incompressible Navier-Stokes equation and modeled in Comsol 6.3.

$$\nabla \cdot \vec{u} = 0$$

$$\frac{\partial \vec{u}}{\partial t} + (\vec{u} \cdot \nabla)\vec{u} = -\frac{1}{\rho}\nabla p + \nu \nabla^2 \vec{u} + \vec{f}$$

Solid obstacles described by no-slip wall conditions obstruct and alter the flow. The boundary conditions at y=0 and y=-1 vary from simulation to simulation and can either be wall, symmetric or periodic. The inlet at x=0 is defined by an inlet velocity. For the wall case, the inlet velocity is parabular-shaped. The system is incompressible and isothermal, yielding zeroed density and temperature fields.

### 6.3.6 HEATED FLOW

Heated flow (Figure 9b) is an extension of the incompressible flow around obstacles. Here, a compressible gas is heated/cooled while flowing through a channel with obstacles. This creates interesting interactions of density-driven convection and the forced convection. Two versions of the systems are used, one with heating rods (top) and one with cooling rods and a heated wall (middle).

Governing equations are the compressible Navier-Stokes equations for conservation of mass, momentum, and energy:

$$\frac{\partial \rho}{\partial t} + \nabla \cdot (\rho \mathbf{u}) = 0 \tag{15}$$

$$\frac{\partial (\rho \mathbf{u})}{\partial t} + \nabla \cdot (\rho \mathbf{u} \otimes \mathbf{u} + p\mathbf{I}) = \nabla \cdot \tau + \rho \mathbf{g} \tag{16}$$

$$\frac{\partial E}{\partial t} + \nabla \cdot ((E + p)\mathbf{u}) = \nabla \cdot (\tau \cdot \mathbf{u} - \mathbf{q}) + \rho \mathbf{g} \cdot \mathbf{u} \tag{17}$$

and the heat conduction equation, which is part of the energy equation above, is described by Fourier's law:

$$\mathbf{q} = -k\nabla T$$

where $\rho$ is the fluid density, $\mathbf{u}$ is the flow velocity vector, $p$ is the pressure, $\mathbf{I}$ is the identity tensor, $\tau$ is the deviatoric stress tensor, $\mathbf{g}$ is the gravitational acceleration, $E$ is the total energy per unit volume, $\mathbf{q}$ is the heat flux vector, $k$ is the thermal conductivity, and $T$ is the temperature.

### 6.3.7 TWOPHASE FLOW

Twophase flow in porous media (Figure 9c) is an important problem in energy systems, hydrology and the petro-industry. In this dataset, water replaces air inside a randomly generated pore structure. For contact angles above 90 degrees (hydrophobic), a positive pressure is applied. For hydrophilic contact angles, a negative pressure is applied. The fluid motion is governed by capillary pressure, surface tension and contact angles. This dataset was generated with COMSOL 6.3 using the phase-field method.

The phase field method describes the interface between immiscible fluids using a continuous dimensionless phase field parameter $\phi$. The system's free energy is given by the functional

$$F(\phi) = \int_\Omega \left( f_{mix}(\phi) + \frac{1}{2}\epsilon^2 |\nabla \phi|^2 \right) dV \tag{18}$$

where $\epsilon$ is a measure of the interface thickness, $f_{mix}$ is the mixing free energy density, and the second term accounts for the energy associated with interface gradients.

The evolution of the phase field parameter, including advection by the velocity field $\mathbf{u}$, is governed by the following equation, which aims to minimize the total free energy density $f_{tot}$ ($J/m^3$) with a relaxation time controlled by the mobility $\gamma$ ($m^3 \cdot s/kg$)

$$\frac{\partial \phi}{\partial t} + \mathbf{u} \cdot \nabla \phi = \nabla \cdot \left( \gamma \nabla \frac{\delta F}{\delta \phi} \right) \tag{19}$$

$$= \nabla \cdot \left( \gamma \nabla \left( \frac{\partial f_{tot}}{\partial \phi} - \epsilon^2 \nabla^2 \phi \right) \right)$$

Here, $f_{tot}$ is the total free energy density, which includes the mixing energy and potentially other contributions like elastic energy.

For an isothermal mixture of two immiscible fluids, the mixing energy density $f_{mix}$ typically assumes the Ginzburg-Landau form:

$$f_{mix}(\phi) = \lambda \left(1 - \phi^2\right)^2 \tag{20}$$

Here, $\phi$ is the dimensionless phase field variable, defined such that the volume fractions of the two fluid components are $(1 + \phi)/2$ and $(1 - \phi)/2$. The quantity $\lambda$ $(N)$ is the mixing energy density, and $\epsilon$ $(m)$ is a capillary width related to the interface thickness. These two parameters are connected to the surface tension coefficient $\sigma$ $(N/m)$ through the equation

$$\sigma = \frac{2\sqrt{2}}{3} \frac{\sqrt{\lambda}}{\epsilon} \tag{21}$$

When considering only mixing energy and gradient energy, the evolution equation (19) simplifies to the Cahn-Hilliard equation:

$$\frac{\partial \phi}{\partial t} + \mathbf{u} \cdot \nabla \phi = \nabla \cdot (\gamma \nabla G) \tag{22}$$

where $G$ $(Pa)$ is the chemical potential, and $\gamma$ $(m^3 \cdot s/kg)$ is the mobility. The mobility controls the timescale of Cahn-Hilliard diffusion and must be chosen appropriately to maintain a constant interfacial thickness without excessively damping convective terms.

The chemical potential $G$ is given by the derivative of the free energy density with respect to the phase field

$$G = \frac{\partial f_{tot}}{\partial \phi} - \epsilon^2 \nabla^2 \phi \tag{23}$$

The Cahn-Hilliard equation drives $\phi$ towards values of $1$ or $-1$ in the bulk phases, with a rapid transition occurring within the thin fluid-fluid interface region. The Phase Field interface in COMSOL Multiphysics typically solves equation (22) by splitting it into two coupled second-order PDEs

$$\frac{\partial \phi}{\partial t} + \mathbf{u} \cdot \nabla \phi = \nabla \cdot (\gamma \nabla G) \tag{24}$$

$$G = \frac{\partial f_{mix}}{\partial \phi} - \epsilon^2 \nabla^2 \phi \tag{25}$$

### 6.3.8 SUPERSONIC FLOW

Supersonic flow is modeled as compressible inviscid flow. The shock front moves with Mach numbers between 1.1 to 5.0. Governing equations are

$$\frac{\partial}{\partial t} \iint_{\Omega} U \, dA + \oint_{\partial \Omega} (F\hat{i} + G\hat{j}) \cdot \hat{n} \, dS = 0 \tag{26}$$

$$\tag{27}$$

where

$$U = \begin{pmatrix} \rho \\ \rho u \\ \rho v \\ \rho E \end{pmatrix} \quad \text{and} \quad F = \begin{pmatrix} \rho u \\ \rho u^2 + p \\ \rho uv \\ u(\rho E + p) \end{pmatrix} \quad G = \begin{pmatrix} \rho v \\ \rho uv \\ \rho v^2 + p \\ v(\rho E + p) \end{pmatrix} \tag{28}$$

Except for the inlet, Neumann boundary conditions are used (sides and outlet). Initial conditions are set to atmospheric conditions (P = 101325 Pa, T = 298 K, $\rho$ = 1.23 kg/m3). The system is isothermal.

## 6.4 KNOWN PHYSICS: DETAILED LONG-HORIZON PREDICTIONS

The following results can be best viewed on a high-definition digital monitor.

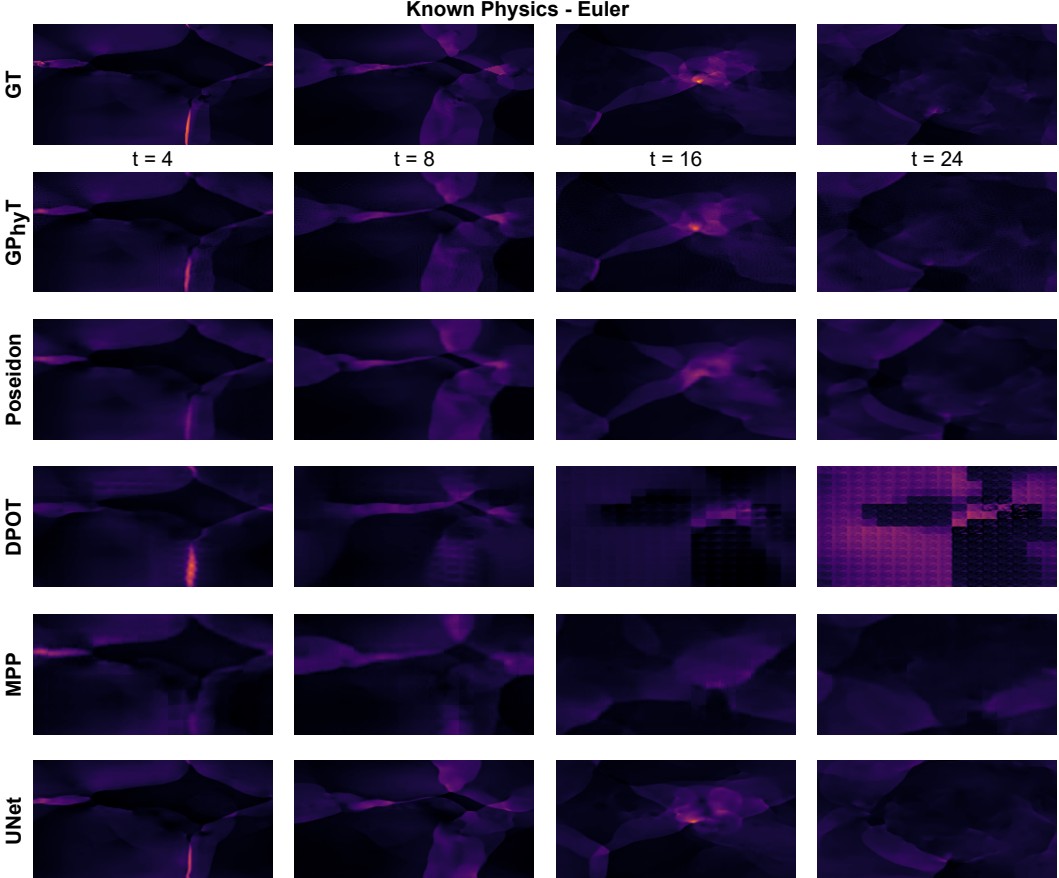

Figure 10: Long-horizon rollouts for all models and ground truth (GT) on the Euler dataset. This is the pressure field with a $\Delta t$ of 1.

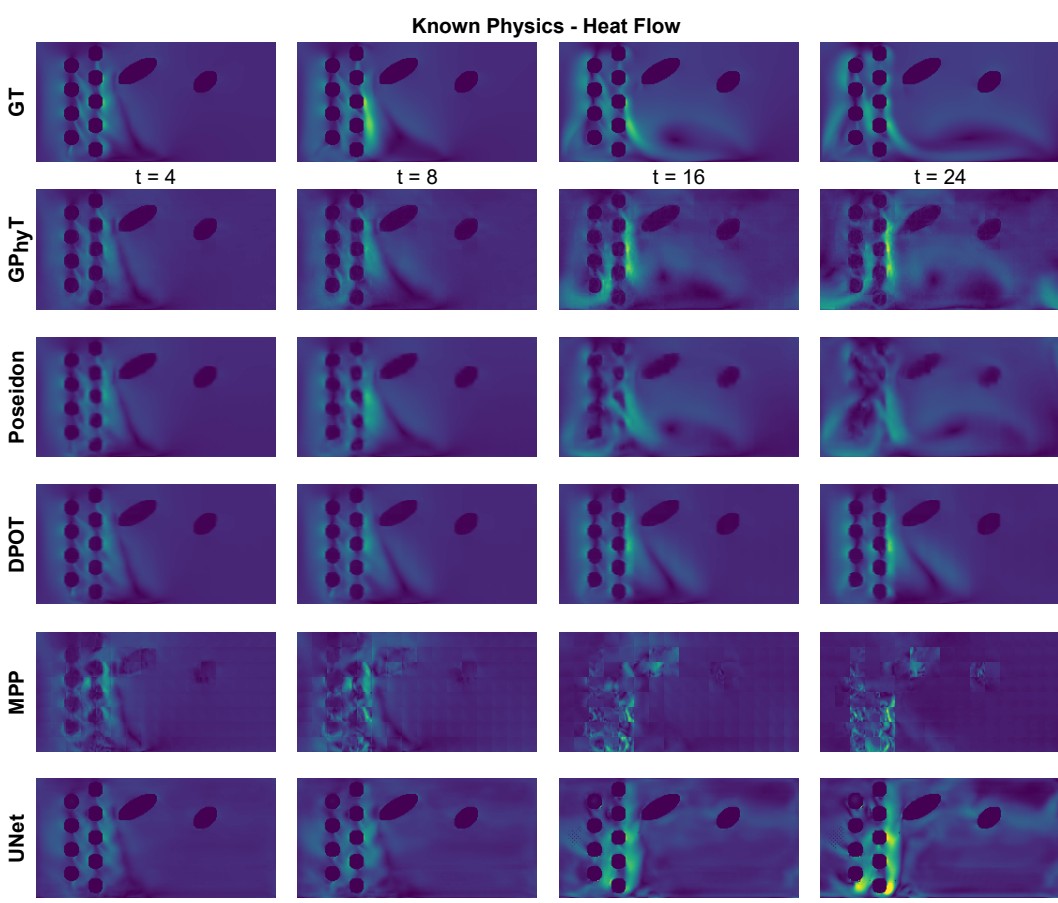

Figure 11: Long-horizon rollouts for all models and ground truth (GT) on the heat-flow dataset. This is the velocity-magnitude (x and y combined) field with a $\Delta t$ of 8.

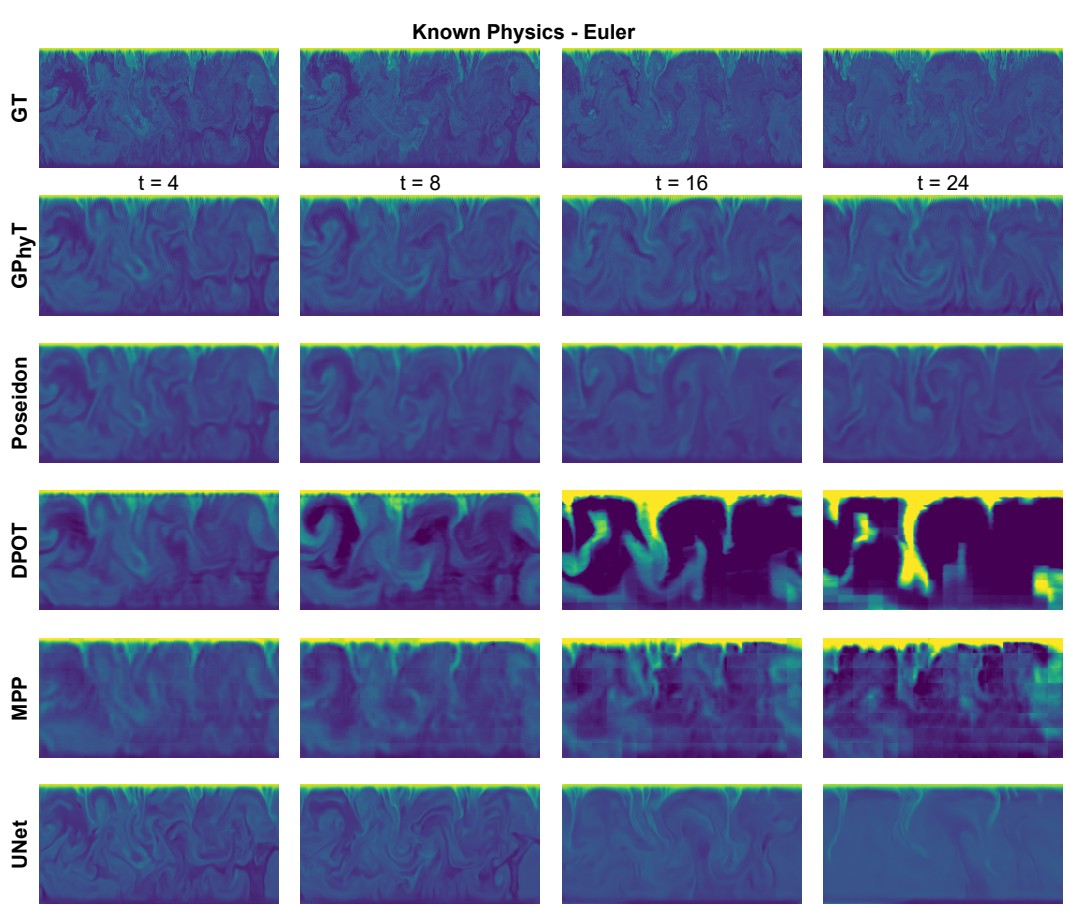

Figure 12: Long-horizon rollouts for all models and ground truth (GT) on the rayleigh-benard dataset. This is the density field with a $\Delta t$ of 1.

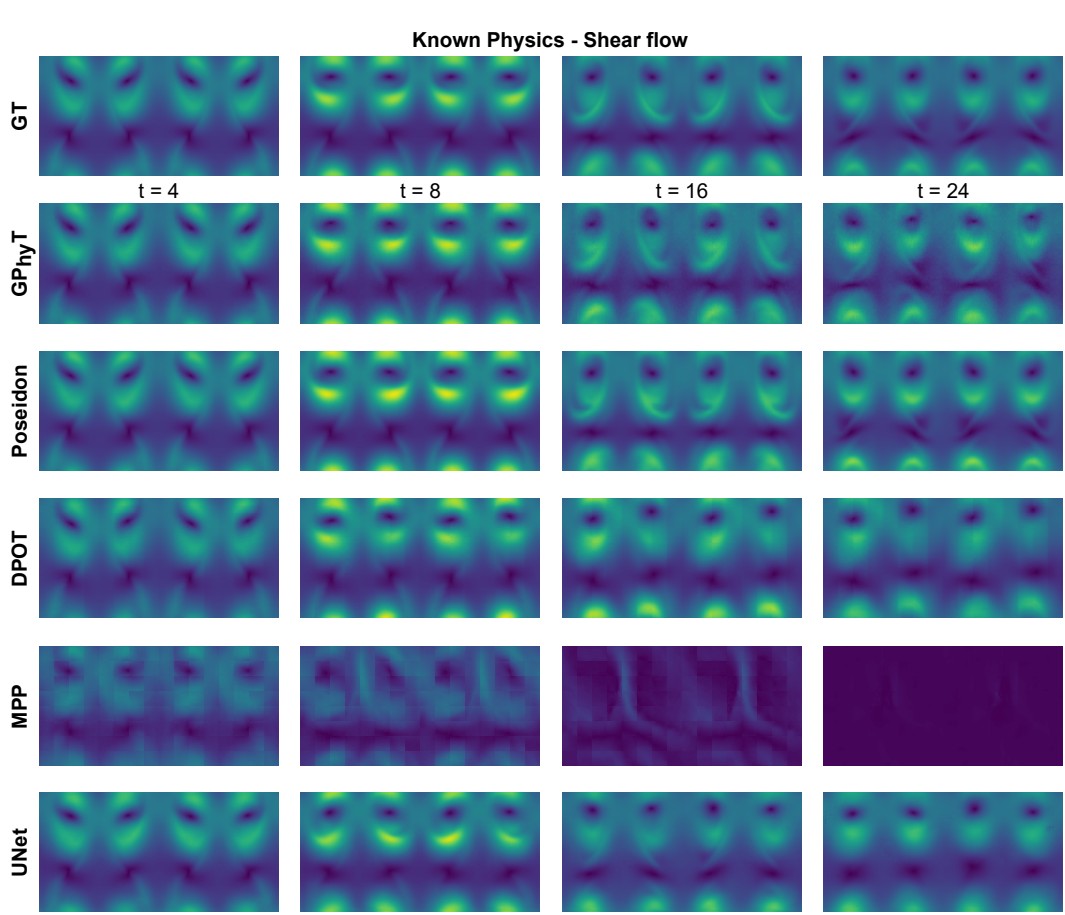

Figure 13: Long-horizon rollouts for all models and ground truth (GT) on the shear flow dataset. This is the velocity-magnitude field with a $\Delta t$ of 1.

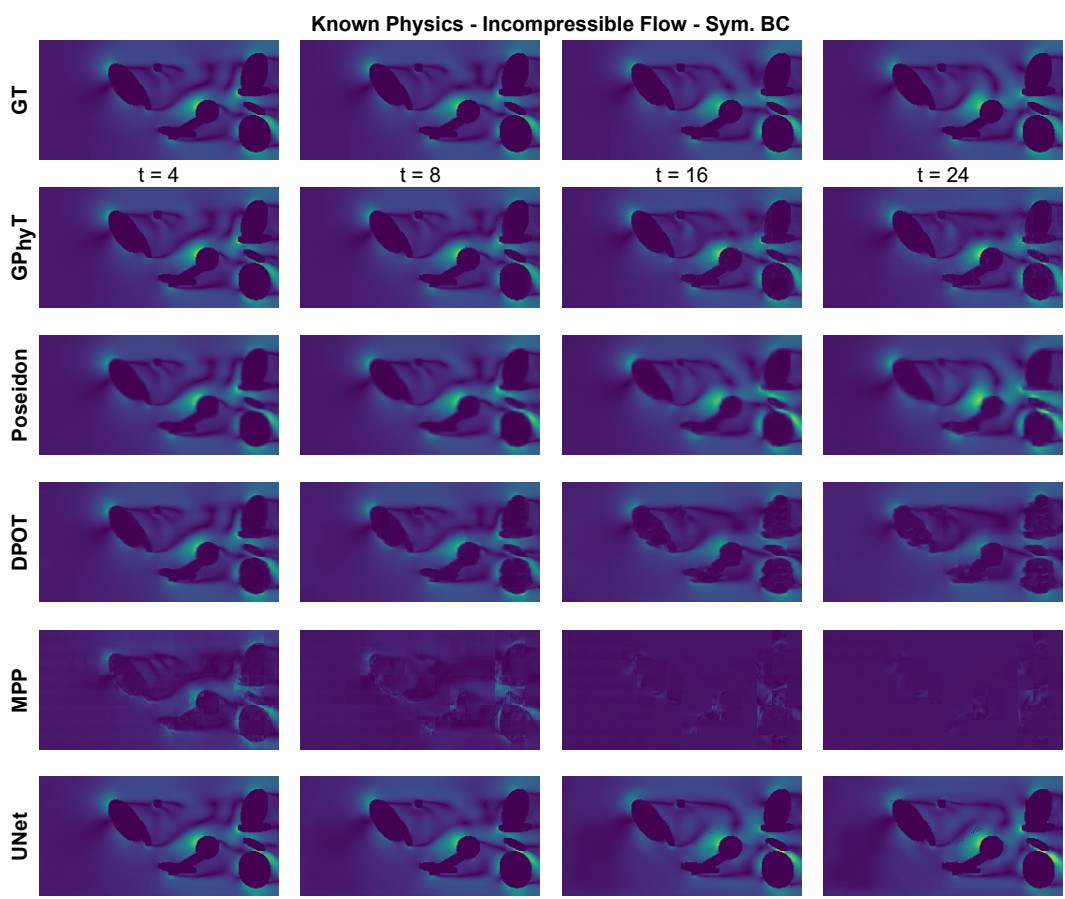

Figure 14: Long-horizon rollouts for all models and ground truth (GT) on the incompressible flow dataset. This is the velocity-magnitude field with a $\Delta t$ of 1.

## 6.5 NOVEL PHYSICS: DETAILED LONG-HORIZON PREDICTIONS

The following results can be best viewed on a high-definition digital monitor.

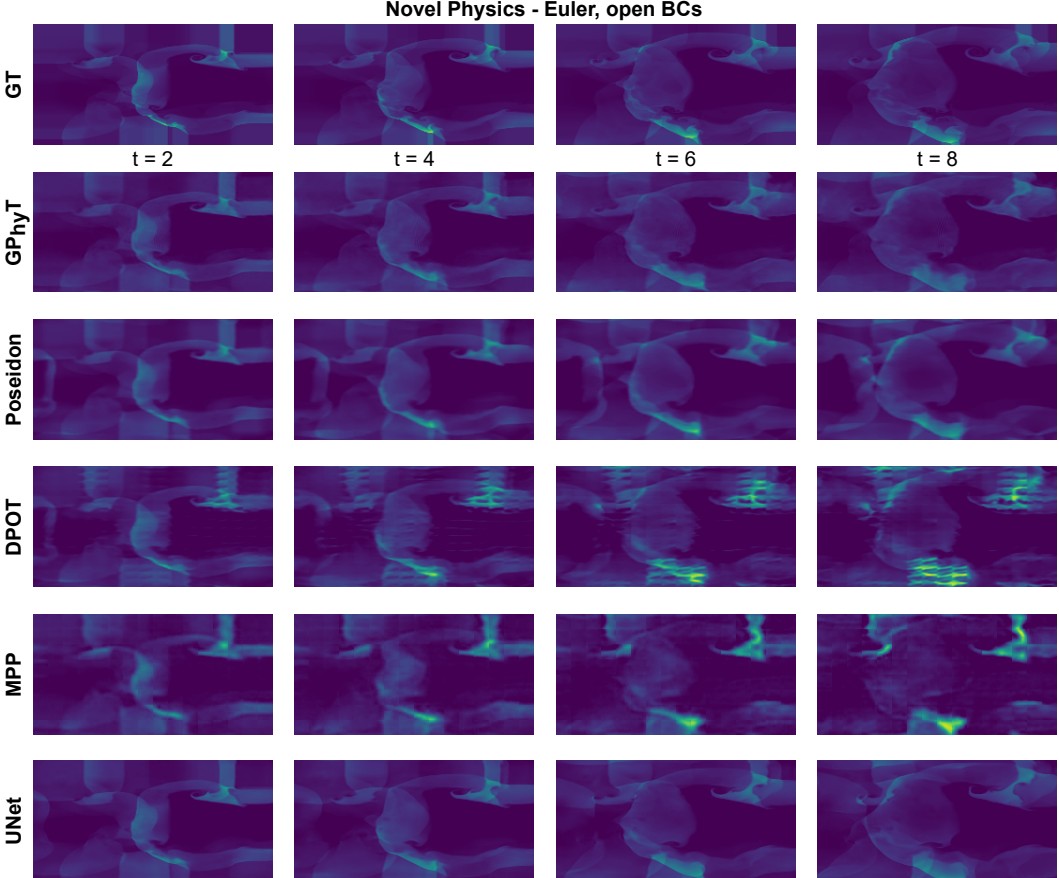

Figure 15: Long-horizon rollout for all models and ground truth (GT) on the novel Euler dataset with open boundary conditions. This is the density field with a $\Delta t$ of 1.

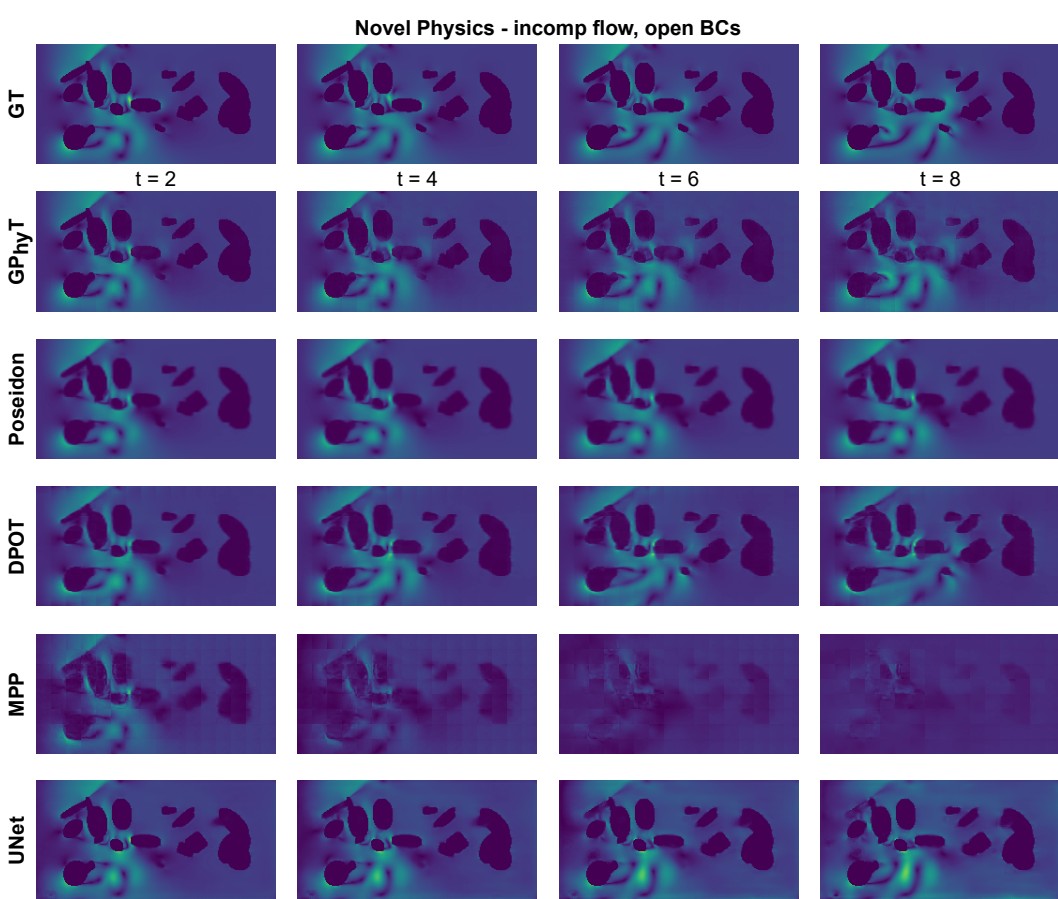

Figure 16: Long-horizon rollout for all models and ground truth (GT) on the novel incompressible flow dataset with open boundary conditions. This is the velocity-magnitude field with a $\Delta t$ of 1.

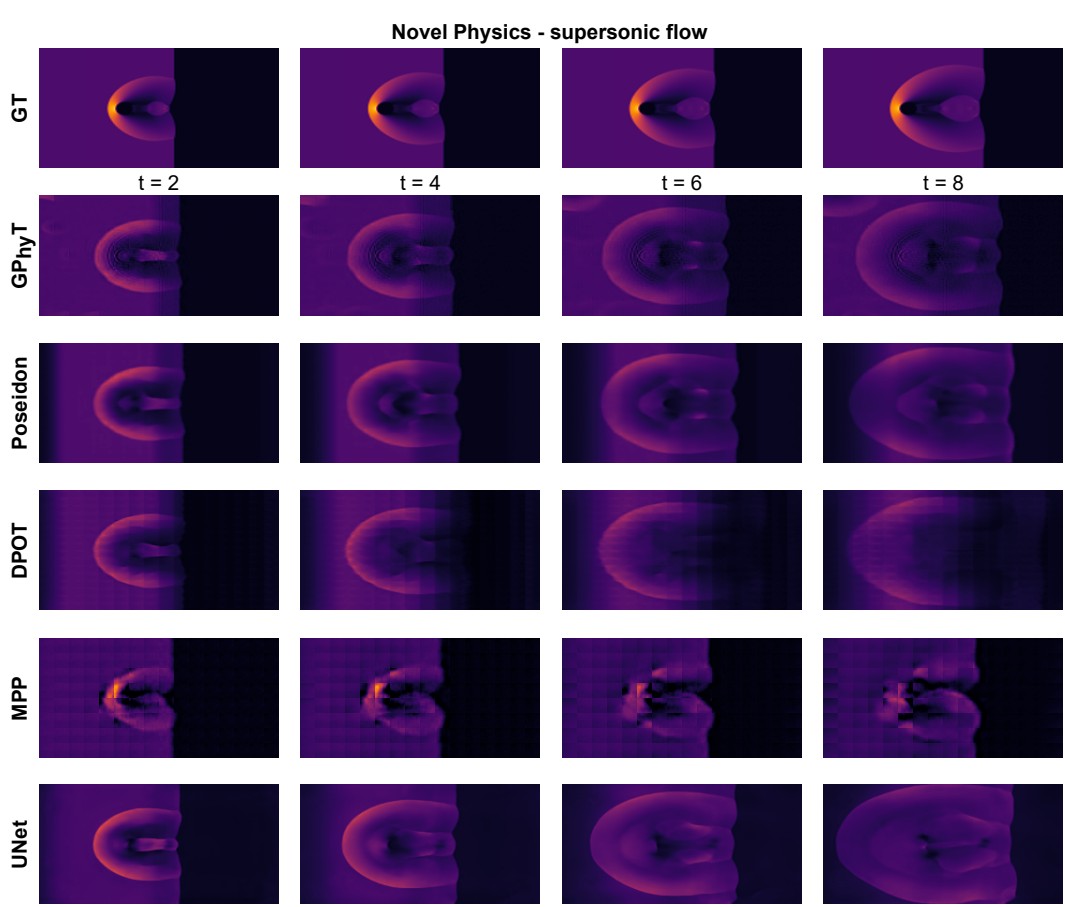

Figure 17: Long-horizon rollout for all models and ground truth (GT) on the supersonic flow dataset. This is the pressure field with a $\Delta t$ of 1.

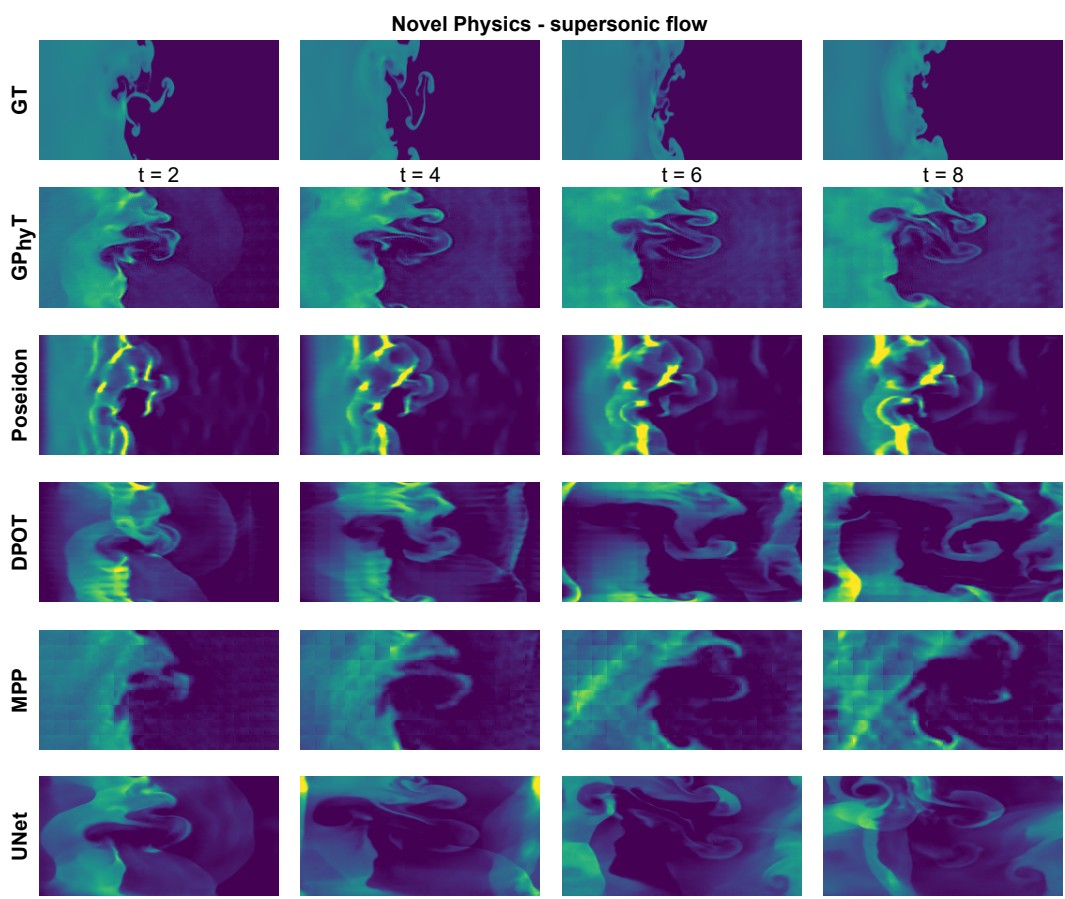

Figure 18: Long-horizon rollout for all models and ground truth (GT) on the novel turbulent radiative layer dataset. This is the density field with a $\Delta t$ of 1.

