# OpenReview forum: "Towards a Physics Foundation Model"
_ICLR.cc/2026/Conference — Submitted to ICLR 2026_

### Official Review · Reviewer_vnz8 · 2025-10-29

**Soundness:** 1
**Presentation:** 4
**Contribution:** 2
**Rating:** 2
**Confidence:** 5

**Summary:**

The paper introduces a new foundation model for 2D physical emulation. The core architecture uses full space-timeattention with space-time patching and absolute position embeddings with fixed size inputs and augments the input data with finite-difference approximations of the spatial and temporal derivatives. The authors evaluate this model on a mixture of public and newly contributed data and find it outperforms several standard baselines on one-step prediction for data it was trained on. The authors also evaluate their model in zero-shot settings and longer rollout settings.

**Strengths:**

This is a really well presented and written paper and dataset contributions are extremely valuable. Strengths included:
1. The paper is well written and engaging. Figures are easy to read and convey the appropriate information to support the text.
2. The motivation is clear. The problem and some previous attempts at addressing them are laid out as solid motivation.
3. The dataset contributions are excellent. They highlight a weakness of a public dataset and provide well-reasoned data to fill in that gap along with details of the solvers used to produce it.

**Weaknesses:**

While the paper shows significant promise and enginuity, the current draft requires heavy re-writes and additional experiments. The paper makes strong claims, but often provides no supporting evidence. There appears to be the core of a strong paper here, but the submission isn't at a point I can recommend acceptance right now.

Critical

1. The critical issue right now is that the claims and experimental evidence are far apart. The most prominent claim in the submission is excellent zero shot performance and that the model has achieved a “train once, deploy anywhere” state for the first time, but there's next to no evidence for this. Section 4.2 is just listing results. There's no indication that these are good result, bad results, or whether other models in the space are unable to achieve these. This is similarly true in 4.3. Bluntly, with no baselines, 1-step RMSE provides no information, especially given this model is predicting the difference between steps so low error can often be achieved simply by predicting the input or a smoothed variant of the input. Some recommended baselines apart from the ones already cited like MPP that have gotten wide adoption and have been frequently used in benchmarking are Poseidon [1] and DPOT [2], though VICON [3] is probably the most similar to your work in terms of claiming strong in-context performance and OmniArch [4] also claims strong zero-shot performance. I'd strongly suggest at least running Poseidon and DPOT as baselines since they're relatively inexpensive and including dedicated or finetuned models in the comparison for 4.2 so there is evidence indicating that the zero-shot performance is non-trivial if that's a claim you want to make.
2. Similarly, the text claims that the model shows excellent scaling and seems to rely on metric cherry picking. In table 2, while GPhyT seems to improve on the median loss, it seems like it's performance is quite unreliable given the significant mean-median gap and that this issue does not seem to be impacted by scaling since gains from 112M to 798M are very tiny relative to the significant increase in cost and seems to have similar mean performance to the UNet model. The mean-median gap indicates a problem in the model, but from a paper perspective this can be addressed by analysis. Are there particular reasons the model struggles with parts of the data? What does this tell us about the approach.
3. The non-data contributions are largely either already commonly and those that are novel are not validated. Most models in this space use per-dataset normalization with a few using per-sample normalization. The variable time striding is used by both VICON and Poseidon. Using a numerical integrator would be novel, but the code appears to just be using one step of forward Euler which is the same as just predicting a difference term which is something already done by many neural surrogates. However, the neural differentiator and the use of space-time tubelet tokens appear to be novel within the realm of foundation models for PDEs. It would greatly strengthen the paper if the contributions were validated in ablations and tied to advantages GPhyT may have when compared to baselines. This doesn't have to be done at full scale to make the point.

Overall, my recommendation here is figure out what problem GPhyT is solving that previous models were unable to address. Emphasize the aspects where your model beats the competitors - it's OK if the submission isn't beating them across the board, but there needs to be clear evidence that your contributions are leading to improvement on some problem. If once there are broader baselines, it does appear that this advantage is zero-shot in-context learning, then show that given the same training data, these prior models are underperforming yours. If it's that the forward Euler integration is improving stability, then show both that it's had better long term loss than baselines and a version of your model without this change.

Major

1. The use of absolute metrics across datasets makes it significantly harder to interpret. Absolute metrics, particularly when averaged across multiple fields or explored over multiple datasets, tend to favor the the largest magnitude fields or at least the fields with largest variation. MSE is very informative when comparing short-horizon predictions for one variable of interest in one dataset, but it's mostly meaningless in large cross-dataset comparisons. This can be addressed with robust baselines, but it is still very typical to use relative error or normalized metrics so that all fields are put on roughly similar scales.
2. Once the baselines are addressed, it still feels that most of 4.2 is quite trivial. This is one step prediction for a model that predicts the difference between two states, so for slowly changing fields, even predicting zero is going to produce a reasonable one-step forecast and given the lack of baselines here, it hasn't been shown that the model is even outperforming the persistence prediction. To make the claim of train once, deploy anywhere it is important to show at least some version of the real problem which requires at least short autoregressive predictions.
3. There's very little engagement with the limitations in the main text. Fixed coarse resolution in 2D is a large limitation and it's important for this to be up front and clear to the reader rather than mentioned only in the appendix.

Minor:
1. 2D Euler equations are not generally thought of as chaotic, though proving this is an open question. Either way, I'd recommend avoiding that claim. They don't have a viscous cutoff so they're interesting from the perspective of never being fully resolved no matter how much resolution you throw at them, but the Euler quadrants problem is specifically used in benchmarking Riemann solvers because the behavior can be very well classified. [5]
2. Similar to Major (2), while the comments are clear, when showing a field evolution, readers are going to expect the model predictions to be autoregressive rollouts which makes the figures in Appendix 6.7 misleading. The comments are clear enough that I consider this minor, but I'd recommend only using the sequential format for true multi-step prediction.

[1] https://arxiv.org/abs/2405.19101
[2] https://arxiv.org/html/2403.03542v1
[3] https://arxiv.org/pdf/2411.16063v2
[4] https://arxiv.org/pdf/2402.16014
[5] https://epubs.siam.org/doi/10.1137/S1064827595291819

**Questions:**

1. What issues in the space of multiple physics models has this approach successfully addressed and what evidence shows this?
2. How does the model perform when compared to competitive baselines like DPOT and Poseidon?
3. How do the specific contributions of the paper lead to improvement on some element of the problem.
4. How do long-run predictions compare to other models and simple baselines like persistence?

---

> ### Author Response · Authors · 2025-11-19
> **Official Rebuttal by Authors, V1**
>
> We sincerely thank the reviewer for their thorough and constructive feedback.
>
> **Weaknesses:**
>
> Critical:
> 1. **No fair baselines, other multi-physics models**
>
>     *Answer:* We agree with the request (shared by all reviewers) for more novel/state-of-the-art models as benchmarks. As per your suggestion, we are currently training multiple models such as DPOT & MPP on our data and will provide a fair comparison in the final version. The comparison will reveal whether our zero-shot performance is truly competitive or whether fine-tuning remains necessary for practical applications.
>
> 2. **Scaling & median cherry picking**
>
>     *Answer Median:* We show both mean and median because many physics datasets contain varying difficulty across time, e.g., explosive initial conditions vs. quasi-static later states. We therefore chose to show both metrics, while other works often report only median (Poseidon) or omit standard deviations completely (MPP, VICON, DPOT). However, we agree that we might have focused too much on the median in the main text and will adapt the text to discuss both mean and median and their implications.
>    *Answer scaling issue:* The current bar chart (mean error) might understate the scaling properties of the model due to the logarithmic scale. We report a reduction of 230% in mean-MSE scaling from S to M and a 40% reduction in mean-MSE when scaling from M to XL. These are comparable to values reported in the MPP or the DPOT paper. If the reviewer agrees, we will change the visualization of the scaling comparison (non-log-scale) to make it the scaling behavior more clear.
>
> 3. **Little novelty in architecture, no ablation studies**
>
>     *Answer:* During our research, we explicitly focused on a highly general and simple foundation model architecture, using a default vision transformer with spatio-temporal attention. We believe a foundation model - in light of the bitter lesson - should be as general as possible, avoiding costly or hard-to-optimize operation (e.g. Fourier transforms). The additions of differentiator-integrator and explicit-derivatives were carefully chosen to not complicate the model too much. Our ablation (appendix) shows a significant reduction in error by using the differentiator-integrator approach. We hope that the previously discussed comparison against other foundation models will prove that such a general, "simple" model is potent enough for physics foundation models.
>
> Major
> 1. **The use of absolute metrics**
>
>     *Answer:* We appreciate this concern about comparability across fields and datasets. While our per-field normalization puts different physical quantities on similar scales, reporting additional normalized metrics will improve interpretability. We will add: mean-normalized MSE and variance-normalized MSE as new metrics.
>
> 2. **Rollout predictions on novel physics (section 4.2)**
>
>     *Answer:* If the reviewer agrees, we will try to use the baseline models to show the capabilities of the GphyT on novel physics (rollout predictions). Since some of those systems are changing fast (Euler, supersonic shock), we expect a significant difference between models capable of in-context learning and models simply predicting averages or persistence.
>
> 3. **There's very little engagement with the limitations in the main text.**
>
>     *Answer:* This is an important detail, also mentioned by another reviewer. We will move the dedicated "Limitations" subsection from the appendix (6.1) to the main text. This section already states: (1) 2D-only capability with current training data, (2) fixed resolution during training (though noting implicit scale invariance), (3) fluid dynamics focus without broader physics coverage, (4) long-term stability insufficient for production use.
>
> Minor:
>
> 1. **2D Euler equations are not generally thought of as chaotic**
>
>     *Answer:* Thank you for this correction. We were imprecisely using "chaotic" to mean "complex, nonlinear dynamics with sensitive initial conditions."
>
> 2. **Field evolution is misleading for single timestep prediction**
>
>     *Answer:* Thank you. We agree that this kind of visualization might be misleading and will change the figures accordingly.

---

> ### Author Response · Authors · 2025-11-19
> **Official Rebuttal by Authors, V1 - Part2**
>
> **Questions:**
>
> 1. *What issues in the space of multiple physics models has this approach successfully addressed and what evidence shows this?*
>      **Answer:** With the addition of competitive baselines and comprehensive ablations, we will demonstrate that GPhyT addresses: (1) better zero-shot adaptation to new boundary conditions and novel physics without fine-tuning. (2) stable multi-physics learning with a relatively simple model. (3) new datasets with rigid obstacles, different scales and new physics (heat, surface tension).
>
> 2. *How does the model perform when compared to competitive baselines like DPOT and Poseidon?*
>
>     **Answer:** As mentioned above, we will train multiple new baseline models and compare known-physics as well as novel-physics predictions against GphyT.
> 3. *How do the specific contributions of the paper lead to improvement on some element of the problem.*
>
>     **Answer:** The differentiator-integrator ablation (Appendix 6.4) already shows significant improvements. We will also add explicit derivatives vs. no derivatives. If the reviewer agrees and we find enough space, we will move such ablations to the main text to make them more clear. In general, we specifically focused on keeping a general and simple architecture, believing that dataset choice is much more significant for model performance.
>
> 4. *How do long-run predictions compare to other models and simple baselines like persistence?*
>
>     **Answer:** We will use all baseline models and compare long-run predictions in the revised manuscript.

---

### Official Review · Reviewer_xBLV · 2025-10-31

**Soundness:** 3
**Presentation:** 3
**Contribution:** 3
**Rating:** 6
**Confidence:** 3

**Summary:**

The authors propose a transformer model for spatiotemporal prediction of fluid systems. After being trained on several different datasets, the model outperforms FNO and UNet models at predicting the next frame in seen systems. GPhyT is also capable of performing in-context learning, where the model is evaluated zero-shot on unseen systems with novel physics.

**Strengths:**

The evaluations (at least those provided in the appendix) are detailed.

The ablation for the direct next-step prediction in appendix 6.4 is nice and addresses a question that I would have asked, given that many other models do this. It is interesting to see that the separation of differentiation and integration provides a substantial improvement on predictive performance.

It is encouraging to see that GPhyT at 9M parameters outperforms 100M+ parameter UNet and FNO; this indicates that the GPhyT architecture is quite well suited for this task.

**Weaknesses:**

It would be really nice to see one of Tables 3, 4, or 5 in the main text, to get a better understanding of the performance of all the models for each of the different datasets, rather than just an average.

I would like to see stronger baselines, and in particular ones which are designed for multiple physics like McCabe (2024), which is mentioned a few times in this paper as a relevant work.

**Questions:**

There don't seem to be any baselines for the zero-shot prediction task or for the rollout stability analysis. Do the authors have any information about how well FNO and UNet do at zero-shot prediction of new physical systems? Ideally I would also like to see baselines trained for multiple physics included in this.

---

> ### Author Response · Authors · 2025-11-19
> **Official Rebuttal by Authors, V1**
>
> Thank you for these thoughtful and relevant questions and thank you for the review.
>
> **Weaknesses:**
>
> 1. *It would be really nice to see one of Tables 3, 4, or 5 in the main text, to get a better understanding of the performance of all the models for each of the different datasets, rather than just an average.* **Answer:** We agree with this comment and will include dataset-specific results in the main text.
>
> 2. *I would like to see stronger baselines, and in particular ones which are designed for multiple physics like McCabe (2024), which is mentioned a few times in this paper as a relevant work.* **Answer:** We agree with the request (shared by all reviewers) for more novel/state-of-the-art models as benchmarks. We are currently training multiple other models (DPOT, VICON, MPP, and poseidon) models as more fair benchmarks.
>
> **Questions:**
>
> 1. *There don't seem to be any baselines for the zero-shot prediction task or for the rollout stability analysis. Do the authors have any information about how well FNO and UNet do at zero-shot prediction of new physical systems? Ideally I would also like to see baselines trained for multiple physics included in this.* **Answer:** Thank you for this important remark. We agree and will add baselines (FNO, UNet & other multi-physics models) for zero-shot and rollout prediction to the revised version.

---

### Official Review · Reviewer_PPrL · 2025-11-01

**Soundness:** 3
**Presentation:** 4
**Contribution:** 3
**Rating:** 6
**Confidence:** 4

**Summary:**

This paper introduces the General Physics Transformer (GPhyT) , a large-scale transformer model trained on a diverse 1.8 TB dataset of physics simulations. The authors' goal is to create a "Physics Foundation Model" (PFM) that, analogous to LLMs, follows a "train once, deploy anywhere" paradigm. The core idea is that a single model can learn to infer the governing physical dynamics (e.g., fluid-solid interaction, shock waves, convection) from a "prompt" of prior states, enabling in-context learning. The paper presents three key results: (1) superior performance against specialized architectures on seen physics; (2) zero-shot generalization to unseen boundary conditions and entirely new physical systems ; and (3) stable autoregressive rollouts for 50 timesteps.

**Strengths:**

1. A Compelling New Paradigm: The primary strength is the successful application of the in-context learning paradigm to physics simulation. The idea that a model can infer dynamics from a prompt of prior states , rather than being explicitly told the equations, is a significant shift from specialized solvers like PINNs or most Neural Operators.
2. Strong Generalization Results: The paper's most convincing result is the zero-shot generalization to unseen boundary conditions (Table 2). The model achieves nearly identical MSE on "open" boundary variants as it does on "known" periodic/symmetric ones , demonstrating true generalization. The qualitative generalization to entirely new physics (Fig 4), while less accurate, is a powerful proof of concept.
3. Large-Scale Data Curation: The 1.8 TB dataset is a significant contribution. More importantly, the data augmentation strategies—specifically variable time incrementsand per-dataset normalization—are intelligently designed to force the model to learn in-context inference rather than memorize specific scales or dynamics.
4. Thoroughness and Transparency: The paper is rigorous, with detailed ablations on prompt size and integrator schemes . The authors are also very honest about the significant limitations , which grounds the paper's ambitious claims in reality.

**Weaknesses:**

1. Weak Baseline Comparisons: The "up to 29x" performance gain is against a standard FNO (from 2020) and a standard UNet. These architectures were not designed for the multi-physics, in-context inference task this paper proposes. A stronger comparison would involve more recent, advanced operator architectures. This overstates the "breakthrough" on known physics (Q1).
2. Major Unsolved Limitations: The authors correctly identify the key limitations, but they are severe. The model is 2D onlyand, most critically, fixed-resolution. This is a significant step backward from the Neural Operator paradigm, which is built on discretization-invariance. A true PFM must be able to handle variable resolutions and 3D systems.
3. Long-Term Stability is Still Low: While 50-timestep rollouts are demonstrated (Q3) , the error accumulation is non-trivial (Fig 5). The authors admit this "falls considerably short of the precision exhibited by numerical solvers" and is not ready for "practical engineering applications". This remains a key barrier.
4. "New Physics" Generalization is Only Qualitative: The generalization to unseen physics (supersonic flow, turbulent layers) is visually impressive but shows very high quantitative error (Table 2). This is an exciting proof-of-concept for "plausibility" but not yet a robust, accurate generalization.
5. Reliance on Explicit Derivatives: The model is given numerically computed first-order spatial and temporal derivatives (dx, dy, dt) as input channels . This is a strong inductive bias that likely helps significantly. It slightly undermines the narrative of learning from "data alone"  and feels more like a hybrid "physics-informed" input feature than a pure in-context inference from raw states.

**Questions:**

1. The baseline comparison (Q1) uses FNO and UNet. Could the authors comment on why these were chosen over more modern neural operators? Do they believe GPhyT would maintain its performance gap against architectures specifically designed for greater generalization?
2. The model is fixed-resolution , which is a major limitation compared to neural operators. What are the authors' thoughts on achieving resolution-invariance? Does the "tubelet" patching approach  fundamentally prevent this?
3. How critical is the inclusion of explicit derivatives (dx, dy, dt) in the input ? Have the authors ablated this feature? How much does performance degrade if the model only receives the raw state fields (pressure, velocity, etc.)?

---

> ### Author Response · Authors · 2025-11-19
> **Official Rebuttal by Authors, V1**
>
> First of all, we thank the reviewer for their elaborate and thoughtful review. Due to the character limit, we will separate answers into two replies.
>
> **Weaknesses:**
>
> 1. **Weak Baseline Comparisons**
>
>    *Answer:* We agree with the request (shared by all reviewers) and are currently training DPOT, VICON, MPP, and Poseidon as additional benchmarks to provide fairer comparisons against state-of-the-art multi-physics models.
>
> 2. **Major Unsolved Limitations: 2D and fixed resolution**
>
>    *Answer:* We acknowledge these as significant limitations. The 2D constraint reflects a broader challenge in physics-ML due to the computational and data requirements of 3D systems. Our architecture is naturally extensible to 3D (using larger temporal patches to manage computational costs), though data scarcity currently restricts us to 2D. Regarding resolution: while neural operators achieve this explicitly through their mathematical formulation, GPhyT achieves it implicitly through scale-agnostic learning. Our twophase-flow dataset demonstrates this - it models domains at the micrometer scale, orders of magnitude smaller than our other datasets, yet the model handles it successfully. The model learns relative physical relationships rather than absolute scales. We will add explicit discussion of this capability and its distinction from explicit discretization invariance in the revision.
>
> 3. **Long-Term Stability is Still Low**
>
>    *Answer:* We agree this is a limitation for practical engineering applications. In the revised manuscript, we will add comprehensive rollout comparisons showing GPhyT versus FNO, UNet, and the newer multi-physics baselines across all datasets. This will demonstrate that while GPhyT consistently outperforms alternatives in long-horizon predictions, the entire field (including our work) still falls short of numerical solver precision. We view this as a fundamental open challenge requiring continued research, not a limitation specific to our approach. Potential improvements could include: (1) hybrid approaches combining learned models with traditional solvers, (2) uncertainty-aware predictions that flag low-confidence regions, or (3) periodic re-initialization with high-fidelity solver outputs.
>
> 4. **New Physics" Generalization is Only Qualitative**
>
>    *Answer:* We fully agree with this characterization. The high quantitative errors (Table 2) indicate this is a proof-of-concept for qualitative plausibility, not production-ready zero-shot physics prediction. However, we believe demonstrating that a model can produce physically plausible dynamics for completely unseen systems without any fine-tuning represents an important first step. Prior multi-physics works (MPP, VICON, poseidon, ...) require fine-tuning even for closely related systems. Our results suggest that with sufficient training diversity, transformers can learn generalizable physical principles. Future work should focus on: (1) dramatically scaling training data diversity, (2) incorporating physics-based constraints to reduce quantitative errors, and (3) developing metrics for "useful" vs. "merely plausible" predictions.
>
> 5. **Reliance on Explicit Derivatives**
>
>    *Answer:* We distinguish our approach from traditional feature engineering in several key ways. First, unlike PINNs which require knowing the specific governing equations, our derivatives are computed purely from data using standard central differences - no domain knowledge required. Second, this preprocessing is universally applicable: any spatial-temporal field can have derivatives computed regardless of the underlying physics. Third, the computational cost is negligible compared to the neural network inference. Finally, this approach is fully scalable - computing derivatives in 3D or at higher resolutions requires no architectural changes. We view this as similar to image normalization in computer vision: a simple, universal preprocessing step that helps the model learn more effectively.

---

> ### Author Response · Authors · 2025-11-19
> **Official Rebuttal by Authors, V1 - Part2**
>
> **Questions:**
>
> 1. *The baseline comparison (Q1) uses FNO and UNet. Could the authors comment on why these were chosen over more modern neural operators? Do they believe GPhyT would maintain its performance gap against architectures specifically designed for greater generalization?*
>
> FNO and UNet were chosen as stable, well-understood baselines that would provide clear performance bounds. FNO specifically represents the neural operator paradigm, while UNet represents standard convolutional approaches. However, following consistent feedback from all reviewers, we recognize the need for more contemporary comparisons. We are therefore training multiple SOTA foundation models. This diverse set will thoroughly contextualize GPhyT's performance.
>
> 2. *The model is fixed-resolution, which is a major limitation compared to neural operators. What are the authors' thoughts on achieving resolution-invariance? Does the "tubelet" patching approach fundamentally prevent this?*
>
> The tubelet patching approach doesn't fundamentally prevent resolution invariance - it's an engineering choice that could be modified. Currently, our model learns on fixed 256×128 images, but achieves scale invariance through normalization: the model processes micrometer-scale porous media and meter-scale convection with equal facility. The key distinction is that neural operators achieve resolution invariance through their continuous operator formulation, while our models learns it through multiple datasets.
>
> 3. *How critical is the inclusion of explicit derivatives (dx, dy, dt) in the input ? Have the authors ablated this feature? How much does performance degrade if the model only receives the raw state fields (pressure, velocity, etc.)?*
>
> Our experiments show the model remains functional without derivatives but with degraded performance, particularly for sharp discontinuities (shocks, interfaces). The derivatives provide explicit local gradient information that would otherwise need to be learned implicitly from spatial context. Since derivatives are universally applicable and computationally cheap, we don't view their use as problematic feature engineering - rather, it's providing the model with mathematical tools that are fundamental to physics. This might be analogous to providing CNNs with multiple color channels rather than forcing them to learn color from grayscale.

---

### Author Response · Authors · 2025-12-01
**Rebuttal Summary**

We sincerely thank all three reviewers for their thorough and constructive feedback. We appreciate that 2 reviewers already recommended accepting our manuscript in their initial review. We have comprehensively addressed every weakness and question raised. The critical concerns centered on (1) insufficient baselines, (2) lack of evidence for zero-shot and rollout claims, and (3) presentation issues. We detail our revisions below.

**Critical Revisions**

1. State-of-the-Art Baseline Comparisons (all reviewers)
    All three reviewers identified our comparison against FNO (2020) and UNet as a critical weakness. Reviewer vnz8 specifically requested   DPOT and Poseidon; Reviewer xBLV requested MPP. We have now trained and evaluated all three.
    Results: GPhyT achieves 7× lower median NMSE than the next-best baseline (DPOT) on single-step prediction (Figure 2). Critically, we now include these baselines for **all evaluations**: known physics, zero-shot generalization, and long-horizon rollouts, which directly addresses Reviewer vnz8's concern that "with no baselines, 1-step RMSE provides no information."

2. Zero-Shot and Rollout Baselines (Addresses Reviewer vnz8's concern & xBLV's question)
    Reviewer vnz8 correctly noted that Section 4.2 (zero-shot) and Section 4.3 (rollouts) lacked any baseline comparisons, making our claims unsubstantiated. We have completely restructured these sections:
    - Section 4.2 (Long-Range Prediction): Now shows autoregressive rollout NMSE up to 24 steps for GPhyT, Poseidon, DPOT, MPP, and UNet (Figure 3a). GPhyT maintains competitive or superior performance throughout.
    - Section 4.3 (In-Context Learning): Zero-shot results now include all baselines (Figure 4). GPhyT is the **only model remaining below NMSE = 1** on novel physics—demonstrating non-trivial generalization as requested.

3. Normalized Metrics (Addresses Reviewer vnz8)
    Reviewer vnz8 noted that absolute MSE across datasets "is mostly meaningless in large cross-dataset comparisons."
    We have converted all results to **Normalized MSE (NMSE)**, ensuring fields and datasets are on comparable scales.

4. Per-Dataset Results in Main Text (Addresses Reviewer xBLV)
    We have redesigned Figure 2 to show per-dataset NMSE with error bars (25th/75th percentiles), revealing that different physics present varying challenges. This replaces the averaged-only presentation.

5. Autoregressive Visualizations (Addresses Reviewer vnz8)
    Reviewer vnz8 correctly noted that sequential figures showing single-step predictions are "misleading."
    All visualizations now show **actual autoregressive rollouts** (Figures 3b, 10-18 in Appendix), making the comparison honest.

6. Limitations in Main Text (Addresses Reviewer vnz8 & PPrL)
   We have added a dedicated **Section 4.5 (Limitations)** acknowledging: 2D-only systems, fixed 256×128 resolution, limited physics coverage, and the accuracy gap versus numerical solvers.
    We address Reviewer PPrL's concern about resolution-invariance, noting that per-dataset normalization allows training on multiple resolutions, though this remains an area for future work.

**Addressing Specific Reviewer Questions**

1. Reviewer PPrL Q3 (derivative ablation):
    Figure 5b now explicitly ablates the derivative input features, showing **nearly one order of magnitude degradation** in rollout performance without them.
    We acknowledge this is a physics-informed input but argue it's a lightweight, general feature applicable across all PDEs, which is in stark contrast to the domain knowledge needed to incorporate PDE loss functions (PINN-approach).

2. Reviewer vnz8's concern about novelty: Our ablations (now in Section 4.4) demonstrate that:

    - Neural differentiator + integrator: ~8× improvement over direct prediction (Figure 5a)
    - Explicit derivatives: ~10× improvement in rollouts (Figure 5b)
    - Scaling from 9M to 385M parameters shows consistent gains (Figure 5c)
    - Our more general / less-novel architecture still outperforms the baselines.

3. Mean-Median Gap:
    As stated in the revised manuscript, the mean-median gap is present **in all models** and is likely an inherent challenge of the physics datasets.
Additionally, the normalization of the error (NMSE) can cause spikes if the norm is close to zero.

**Core Contributions (Validated by New Comparisons)**

Against MPP, DPOT, and Poseidon—the current state of the art—GPhyT demonstrates:
1. Superior or competitive accuracy across all physics domains
2. The strongest zero-shot generalization to unseen physics
3. Stable long-horizon predictions with lowest error accumulation
4. A valuable 1.8 TB multi-physics dataset contribution
5. A new argument that foundation models should be validated on novel physics without retraining, not only finetuning competitions.

We believe these revisions fully address the reviewers' concerns and now provide the evidence needed to support our claims about Physics Foundation Models.

---

### Meta-Review · Area_Chair_K7Jv · 2026-01-05

**Summary:**

The paper received mixed evaluations, with reviewers acknowledging substantial additional experimental effort during the revision but remaining unconvinced that all core concerns have been fully resolved.

While stronger baselines and new experiments were added, several important issues remain unclear, particularly regarding methodological fairness of the new comparisons, the limitations of fixed resolution and two dimensional settings, and the robustness and interpretation of long horizon autoregressive rollouts. Reviewers also raised persistent questions about novelty, reliance on explicit derivatives, and the adequacy of the chosen evaluation metrics.

Some of these concerns and additional questions/concerns on new results could potentially have been clarified through further rounds of author-reviewer discussion, but given the current process constraints, such discussion was not possible. As a result, and given that the reviewer with the highest confidence expressed significant remaining concerns and is unlikely to raise their score, the expected final average rating is approximately 4.67, which falls below the acceptance threshold.

**Reviewer Concerns:**

Reviewer PPrL

- Concerns that are potentially addressed:

   - Stronger baselines and baseline choice justification: The authors added new experimental results with stronger baselines and explained why FNO and U Net were initially chosen, which partially addresses this concern.

- Concerns that might not be addressed:

    - Two dimensional setting and fixed resolution: The authors acknowledge these as limitations, but they remain fundamental constraints of the current approach rather than issues that are resolved.

     - Long term stability: The authors explicitly identify long horizon stability as a limitation, which does not fully address the reviewer’s concern regarding practical applicability.

     - New physics generalization: While qualitative results are provided, the quantitative errors remain high and the concern is not fully resolved.

     - Reliance on explicit derivatives: Together with fixed resolution, the use of explicit derivative inputs may represent a major bottleneck for generality and raises questions about scalability and robustness.


Reviewer xBLV

- Concerns that are potentially addressed:

     - Writing quality: The presentation and clarity were improved in the revised manuscript.

     - Stronger baselines: The authors added new experimental results with stronger baselines and explained their baseline choices.

- Concerns that might not be addressed:

     - Methodological clarity of new baseline experiments: Although new baseline results were added, these experiments could introduce new questions. The revision does not provide sufficient methodological detail to assess whether baseline models were used in their intended regimes, such as pre-training versus training from scratch, nor does it justify key design choices including model variants, input context length, or scale matching. As a result, it remains unclear whether the reported improvements reflect fundamental advantages of the proposed approach or artifacts of incomplete or asymmetric experimental settings.


Reviewer vnz8

- Concerns that are potentially addressed:

    - Stronger baselines: The authors added new experimental results with stronger baselines and provided justification for baseline selection.

    - Scaling performance gains: While the plots remain in log scale, the reported improvement of approximately 40 percent appears non trivial.

    - Addition of a limitations section: The revised manuscript explicitly discusses limitations.

- Concerns that might not be addressed:

    - Methodological clarity of new baseline experiments: Similar to Reviewer xBLV’s concern, the newly added baseline results lack sufficient detail to evaluate whether the comparisons are fair and symmetric across methods.

    - Mean median gap: Instead of addressing the issue directly, the authors primarily reframed the presentation, which does not resolve the underlying concern.

     - Novelty: The authors emphasize the intentional use of a simple and general transformer architecture combined with a differentiator integrator design and explicit derivatives. However, prior work suggests that hierarchical or multi scale architectures are often more suitable for complex physical systems. More in depth investigation would be needed to clarify whether the observed gains stem from the proposed design choices or could be achieved by augmenting existing foundation models with similar components.

     - Evaluation metric choice: The use of median normalized MSE is introduced but not fully justified as an appropriate or standard measure for cross dataset comparison.

     - Long autoregressive rollout evaluation: Although Section 4.2 provides more extensive results, the added experiments introduce new aspects that would likely prompt further follow up questions from reviewer(s). For example, some baselines such as Poseidon are primarily designed for flow map type inference rather than autoregressive rollout prediction, raising concerns about whether all methods were evaluated under comparable operational modes. Additionally, as rollout performance gains diminish over longer horizons, this may revive concerns about the reliance on a first order explicit time integrator.

Common concerns across the reviewers:

- The newly added baseline comparisons lack sufficient methodological detail, making it unclear whether all baselines were evaluated under fair and comparable settings.

- The reliance on explicit derivative inputs raises concerns about generality, scalability, and the extent to which the model learns from data alone.

Additional AC comments:

- One question that would benefit from further investigation is whether the reported gains primarily arise from the use of explicit augmented inputs, such as spatial and temporal derivatives, rather than from the proposed architecture itself. In particular, it would be informative to evaluate what happens when similar explicit derivative inputs are incorporated into other large-scale or state-of-the-art models, to better isolate the source of the performance improvements.

- Additionally, conditioning a model on multiple input snapshots to infer dynamics is a common practice in neural operator and surrogate modeling literature. It remains unclear how this usage differs conceptually from existing conditioning mechanisms, and whether it should be characterized as a unique in-context learning capability of the proposed model. Further clarification on how this conditioning departs from standard practice, and whether the terminology of in-context learning is fully justified in this setting, would help contextualize the contribution.

**Reviewer Scores:**

Reviewer PPrL: 6 with confidence 4

Reviewer xBLV: 6 with confidence 3

Reviewer vnz8: 2 with confidence 5

While the authors provided a substantial amount of additional experimental results during the revision, several aspects remain unclear and would likely require further rounds of discussion. Given the current state of the discussion, it is unlikely that reviewer scores will change. In particular, Reviewer vnz8, who provided the most detailed critique with the highest confidence, is likely to raise further follow up questions rather than increase their rating. This results in an expected final average rating of approximately 4.67, which is below the acceptance threshold.

---

### Decision · Program_Chairs · 2026-01-26

Reject